# RETHINKING ADVERSARIAL TRANSFERABILITY FROM A DATA DISTRIBUTION PERSPECTIVE

**Yao Zhu**[1]*, **Jiacheng Sun**[2]†, **Zhenguo Li**[2]
[1]Zhejiang University, [2]Huawei Noah's Ark Lab
ee_zhuy@zju.edu.cn, {sunjiacheng1,li.zhenguo}@huawei.com

## ABSTRACT

Adversarial transferability enables attackers to generate adversarial examples from the source model to attack the target model, which has raised security concerns about the deployment of DNNs in practice. In this paper, we rethink adversarial transferability from a data distribution perspective and further enhance transferability by score matching based optimization. We identify that some samples with injecting small Gaussian noise can fool different target models, and their adversarial examples under different source models have much stronger transferability. We hypothesize that these samples are in the low-density region of the ground truth distribution where models are not well trained. To improve the attack success rate of adversarial examples, we match the adversarial attacks with the directions which effectively decrease the ground truth density. We propose Intrinsic Adversarial Attack (**IAA**), which smooths the activation function and decreases the impact of the later layers of a given normal model, to increase the alignment of adversarial attack and the gradient of joint data distribution. We conduct comprehensive transferable attacks against multiple DNNs and show that our **IAA** can boost the transferability of the crafted attacks in all cases and go beyond state-of-the-art methods.

## 1 INTRODUCTION

Deep neural networks (DNNs) are widely used in various safety-critical fields, but they are vulnerable to adversarial examples (Szegedy et al., 2013). Adversarial attacks are imperceptible to humans but catastrophic for the DNNs and can be transferred between different models (Goodfellow et al., 2015; Liu et al., 2017). Adversarial transferability enables attackers to generate adversarial examples from the source model to attack unknown target models, which has raised security concerns about the deployment of DNNs in practice. Understanding the essence of adversarial transferability is a fundamental problem in deep learning. On the one hand, some works show that the characteristics of the source model, such as model architecture (Wu et al., 2019), model capacity (Tramèr et al., 2017), and test accuracy (Wu & Zhu, 2020), influence adversarial examples' transferability. On the other hand, some works think that the data-relevant information may be the key factor for adversarial transferability. Ilyas et al. (2019) explain that adversarial perturbations are non-robust features and not meaningless bugs, but it is hard to specifically define non-robust features. We want to further study transferability quantitatively from the data distribution perspective.

It has been empirically observed that DNNs are relatively robust to random noise (Fawzi et al., 2016). However, in this work we find an intriguing phenomenon: some samples are sensitive to Gaussian noise, in the sense that injecting small Gaussian noise into these samples can fool different models trained on the same dataset. Furthermore, their adversarial counterparts generated by different source models have much stronger transferability against different target models than other samples. We hypothesize that these samples are in the low-density regions of the ground truth distribution both source and target models are trained on, and models are not well trained in these regions. Thus predictions of these samples are easy to be perturbed and even not robust to small random noises. We denote this kind of data as Low-Density Data (LDD), while others as High-Density

---

*This work was done when Yao Zhu was a research intern in Huawei Noah's Ark Lab.
†Corresponding to: Jiacheng Sun

Data (HDD). As shown in Fig. 1 (Left), the attack success rate against different target models of LDD with different strengths of Gaussian noise is much higher than that of HDD. Furthermore, in Fig. 1 (Right), the adversarial counterparts of LDD have much stronger transferability than the adversarial counterparts of HDD (see Appendix B for details).

This phenomenon reveals that the location of data plays a vital role in adversarial transferability and the adversarial examples of samples in the low-density region are strongly transferable. The most efficient direction towards the low-density region is $-\nabla_{\boldsymbol{x}} \log p_D(\boldsymbol{x}, y)$, where $p_D(\boldsymbol{x}, y)$ is the ground truth density of natural data. We name this direction **Intrinsic Attack** because it doesn't depend on the models and only depends on the ground truth distribution. Thus, we propose to match the adversarial attack with intrinsic attack for generating strong transferable adversarial examples.

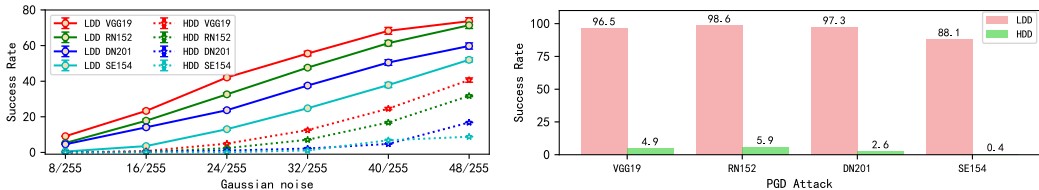

Figure 1: (Left) The attack success rate when injecting Gaussian noise into LDD and HDD. See Appendix L for more corruption experiments. (Right) The attack success rate of the adversarial examples for LDD and HDD by PGD ($\ell_{\infty}$, $\epsilon$=16/255) against different target models (VGG19, RN152, DN201, SE154).[1] The source model is ResNet-50.

We explore the potential of a classifier $p_{\theta,\Lambda}(y|\boldsymbol{x})$ with parameters $\theta$ and structure hyper-parameters $\Lambda$ (see Sec. 3.1) to generate more transferable adversarial examples by aligning adversarial attack with intrinsic attack $-\nabla_{\boldsymbol{x}} \log p_D(\boldsymbol{x}, y)$. The adversarial attack of $p_{\theta,\Lambda}(y|\boldsymbol{x})$ is usually generated by PGD/FGSM method, and is determined by $-\nabla_{\boldsymbol{x}} \log p_{\theta,\Lambda}(y|\boldsymbol{x})$. We match the **A**lignment between the **A**dversarial attack and **I**ntrinsic attack (**AAI**), $\mathbb{E}_{p_D(\boldsymbol{x},y)} \left[ \frac{\nabla_{\boldsymbol{x}} \log p_{\theta,\Lambda}(y|\boldsymbol{x})}{\|\nabla_{\boldsymbol{x}} \log p_{\theta,\Lambda}(y|\boldsymbol{x})\|_2} \cdot \nabla_{\boldsymbol{x}} \log p_D(\boldsymbol{x}, y) \right]$, by modifying the structure parameters $\Lambda$ for a pre-trained network.

In order to maximize **AAI**, we should make $p_{\theta,\Lambda}(y|\boldsymbol{x})$ smoother. Otherwise, $\nabla_{\boldsymbol{x}} \log p_{\theta,\Lambda}(y|\boldsymbol{x})$ will oscillate frequently and hard to match $\nabla_{\boldsymbol{x}} \log p_D(\boldsymbol{x}, y)$. For the commonly used ReLU network, we can smooth it by replacing ReLU activation with Softplus (Nair & Hinton, 2010) with little change of the model's output. Maennel et al. (2020) show that the early layers of a network learn the local statistics of the data distribution better than the later layers, which motivates us to decrease the impact of later layers when generating adversarial examples to utilize the data distribution-relevant information. We can closely match the adversarial attack with the intrinsic attack $-\nabla_{\boldsymbol{x}} \log p_D(\boldsymbol{x}, y)$ and improve the adversarial transferability by optimizing structure hyper-parameters $\Lambda$ to maximize **AAI** as the objective function. We name our method as **Intrinsic Adversarial Attack (IAA)**.

There are some interesting observations in our **IAA** experiments. Firstly, we find that the test accuracy of the source model may not be important. As shown in Fig 2, the accuracy of the pre-trained model with $Softplus_{\beta=15}$ is around 60%, but the adversarial transferability of this model is much stronger than the model with $Softplus_{\beta=45}$. Secondly, although the existing methods (Madry et al., 2018; Wu et al., 2019) can significantly decrease the top-1 accuracy of the target models, the top-5 accuracy is still high. **IAA** can both decrease the top-1 accuracy and top-5 accuracy. Furthermore, the existing methods (Xie et al., 2019; Wu et al., 2019) can just slightly improve the one-step attack under different strengths, while our **IAA** surpasses the existing methods by a large margin. These phenomena verify our hypothesis that **IAA** pulls examples to the low-density region, which causes prediction difficulty to the target models. Our main contributions are summarized below:

- We propose an effective metric, **AAI**, to evaluate the alignment of the model's adversarial attack with intrinsic attack $-\nabla_{\boldsymbol{x}} \log p_D(\boldsymbol{x}, y)$. Furthermore, we show that **AAI** is also an effective metric for adversarial transferability.

---

[1]The **AAI** metric for LDD (0.1264) is much larger than HDD (0.0457), which shows that the direction of PGD attack on LDD aligns better than that on HDD (The **AAI** metric on all test samples is 0.052).

- We propose the Intrinsic Adversarial Attack (**IAA**) by maximizing **AAI** to generate more transferable adversarial examples.
- We conduct comprehensive transfer attack experiments from different source models against nine naturally trained models and three ensemble secured models, showing that **IAA** can significantly improve the state-of-the-art transferability (both targeted and untargeted attack) of adversarial examples (even improve 20% under some settings).

## 2 RELATED WORK

There are two types of adversarial attacks: white-box attacks and black-box attacks. White-box attacks assume that the attacker can completely access the structure and parameters of the target model. Typical examples of white-box attacks are FGSM (Goodfellow et al., 2015), PGD (Madry et al., 2018), and CW (Carlini & Wagner, 2017). The black-box attack assumes that the attacker only knows the output of the target model (prediction or confidence). Black-box attacks are roughly divided into two types: estimating gradient with queries to the target model (Papernot et al., 2017; Su et al., 2019; Yang et al., 2020) and attacking a surrogate model (Xie et al., 2019; Dong et al., 2018; Wu et al., 2019). Attacking a surrogate model is much more efficient and can reduce the risk of exposure. Thus, many existing works focused on adversarial transferability.

Su et al. (2018) explore the factors influencing the transferability and show the architecture has greater influence than model capacity. Dong et al. (2018) show that the momentum of gradients can be used to improve the adversarial transferability. Xie et al. (2019) show the diversity of input data will enhance the adversarial transferability. Huang et al. (2019) fine-tune the adversarial examples by increasing perturbation on a pre-specified layer. Wang et al. (2020) propose a loss to decrease interactions between perturbation units during attacking. Wu et al. (2019) propose that reducing gradients from the residual modules is effective for improving transferability. Guo et al. (2020) removes ReLU activations in the later layers to get linear backpropagation and decreases the influence of intermediate layers. They only modify the backpropagation when generating adversarial examples while keeping the forward prediction as the original model. Based on this, Zhang et al. (2021) conjecture that backpropagating smoothly might be sufficient for improving transferability.

There are also some works on adversarial attack and defense using generative models. Naseer et al. (2019) and Yang et al. (2021a) learn adversarial perturbation through a conditional generative attacking model, which needs to be carefully designed for certain classes. Samangouei et al. (2018); Song et al. (2018) use GANs or autoregressive models to detect and purify adversarial examples. Du & Mordatch (2019); Hill et al. (2021); Srinivasan et al. (2021); Yoon et al. (2021) purify adversarial examples by EBM or score-based generative models. JEM (Grathwohl et al., 2020) shows that combining a classifier with EBM can help to obtain some robustness. However, adversarial attacks or purification based on generative models are computationally costly. We want to modify a normal classifier with little computation cost to enhance its adversarial transferability by maximizing our **AAI** metric.

## 3 METHODS

### 3.1 ALIGNMENT BETWEEN THE ADVERSARIAL ATTACK AND INTRINSIC ATTACK

For a classifier $f_{\theta,\Lambda}$ parameterized by $\theta$ with structure hyper-parameters $\Lambda$ (e.g., hyper-parameters for architecture, activation function, etc.), and data $\boldsymbol{x}$, label $y$, total possible classes $n$, then $f_{\theta,\Lambda}(\boldsymbol{x})[k]$ represents the $k^{th}$ output of the last layer. The conditional density $p_{\theta,\Lambda}(y|\boldsymbol{x})$ can be expressed as:

$$p_{\theta,\Lambda}(y|\boldsymbol{x}) = \frac{\exp(f_{\theta,\Lambda}(\boldsymbol{x})[y])}{\sum_{k=1}^{n} \exp(f_{\theta,\Lambda}(\boldsymbol{x})[k])}. \tag{1}$$

The adversarial attack is usually based on $-\nabla_{\boldsymbol{x}} \log p_{\theta,\Lambda}(y|\boldsymbol{x})$ (Madry et al., 2018; Goodfellow et al., 2015). The most effective direction towards low-density region is intrinsic attack $-\nabla_{\boldsymbol{x}} \log p_D(\boldsymbol{x}, y)$. To improve the adversarial transferability, we need to match model's adversarial attack direction with intrinsic attack. We define the inner product of normalized $-\nabla_{\boldsymbol{x}} \log p_{\theta,\Lambda}(y|\boldsymbol{x})$ and $-\nabla_{\boldsymbol{x}} \log p_D(\boldsymbol{x}, y)$ to quantify the matching of the direction of adversarial attack and the intrinsic attack.

**Definition 1** (**AAI**). For a classifier $p_{\theta,\Lambda}(y|\boldsymbol{x})$, the **A**lignment between its **A**dversarial attack and the **I**ntrinsic attack is:

$$\mathbf{AAI} \triangleq \mathbb{E}_{p_D(\boldsymbol{x},y)} \left[ \frac{\nabla_{\boldsymbol{x}} \log p_{\theta,\Lambda}(y|\boldsymbol{x})}{\|\nabla_{\boldsymbol{x}} \log p_{\theta,\Lambda}(y|\boldsymbol{x})\|_2} \cdot \nabla_{\boldsymbol{x}} \log p_D(\boldsymbol{x},y) \right], \tag{2}$$

where $p_D(\boldsymbol{x},y)$ is the ground truth joint distribution.

*Remark.* (1) We use the normalized adversarial attack to remove the influence of scaling factor when comparing different models.
(2) This definition is equivalent with a modified score matching objective as:

$$\frac{1}{2} \mathbb{E}_{p_D(\boldsymbol{x},y)} \left\| \frac{\nabla_{\boldsymbol{x}} \log p_{\theta,\Lambda}(y|\boldsymbol{x})}{\|\nabla_{\boldsymbol{x}} \log p_{\theta,\Lambda}(y|\boldsymbol{x})\|_2} - \nabla_{\boldsymbol{x}} \log p_D(\boldsymbol{x},y) \right\|_2^2 = -\mathbf{AAI} + C_{p_D},$$

where $C_{p_D}$ is a constant only depend on the ground truth distribution $p_D$.

As getting the gradient of $p_D(\boldsymbol{x},y)$ is not feasible, we use integration by parts (Hyvärinen & Dayan, 2005) to move the gradient on $p_D(\boldsymbol{x},y)$ to model's adversarial attack. With the smoothness assumption on $\nabla_{\boldsymbol{x}} p_{\theta,\Lambda}(y|\boldsymbol{x})$, we have the following theorem:

**Theorem 1.** *If $\nabla_{\boldsymbol{x}} p_{\theta,\Lambda}(y|\boldsymbol{x})$ is differentiable almost everywhere, then*

$$\boldsymbol{AAI} = -\mathbb{E}_{p_D(\boldsymbol{x},y)} \left[ \nabla_{\boldsymbol{x}} \cdot \frac{\nabla_{\boldsymbol{x}} p_{\theta,\Lambda}(y|\boldsymbol{x})}{\|\nabla_{\boldsymbol{x}} p_{\theta,\Lambda}(y|\boldsymbol{x})\|_2} \right]. \tag{3}$$

*Moreover,*

$$\boldsymbol{AAI} = -\mathbb{E}_{p_D(\boldsymbol{x},y)} \mathbb{E}_{p(\boldsymbol{v})} \left[ \boldsymbol{v}^{\mathrm{T}} \nabla_{\boldsymbol{x}} \frac{\boldsymbol{v}^{\mathrm{T}} \nabla_{\boldsymbol{x}} p_{\theta,\Lambda}(y|\boldsymbol{x})}{\|\nabla_{\boldsymbol{x}} p_{\theta,\Lambda}(y|\boldsymbol{x})\|_2} \right], \tag{4}$$

*where $p(\boldsymbol{v})$ is a distribution of random vector v such that $\mathbb{E}_{p(\boldsymbol{v})}[\boldsymbol{v}\boldsymbol{v}^{\mathrm{T}}] = I$ (e.g., the multivariate standard normal $\mathcal{N}(\boldsymbol{0}, I)$).*

This theorem makes it possible to calculate **AAI** without knowing the gradient of ground truth distribution when the model is smooth, see Appendix E for the proof. Combined with sliced score matching (Song et al., 2020) we can efficiently approximate **AAI** on discrete samples. However, for ReLU networks, the model's second derivative is not well defined, which prevents us from using this theorem. Thus, we need to smooth the ReLU models for better modifying the model to maximize **AAI** and improve the transferability. One obvious way to improve the smoothness is to replace ReLU activation with some smooth activation. In this paper, we use Softplus to show that smoothness can greatly help us to improve transferability.

## 3.2 INTRINSIC ADVERSARIAL ATTACK FOR NORMAL MODELS

### 3.2.1 SMOOTHING THE CLASSIFIER BY SOFTPLUS

Srinivas & Fleuret (2021) claim that the input-gradient of a classifier is somewhat aligned with the gradient of the ground truth data distribution $p_D(\boldsymbol{x}, y)$. On the other hand, Dombrowski et al. (2019) show that the information in gradients may be hidden for that the gradient of the ReLU activation function is discontinuous and will frequently change when the input closes to zero. Thus better gradient with less noise can contribute to matching the input-gradients $-\nabla_{\boldsymbol{x}} \log p_{\theta,\Lambda}(y|\boldsymbol{x})$ and the gradients of ground truth distribution $-\nabla_{\boldsymbol{x}} \log p_D(\boldsymbol{x}, y)$. We provide a simple implementation that replacing the widely used activation function ReLU with $\text{Softplus}_\beta$ (Nair & Hinton, 2010):

$$\text{Softplus}_\beta(x) = \frac{1}{\beta} log(1 + \exp(\beta x)), \tag{5}$$

where $\beta$ is the shape-related hyper-parameter. The larger $\beta$ is, the better $\text{Softplus}_\beta$ approximates ReLU. The $\text{Softplus}_\beta$ can also be expressed as the expectation of ReLU in a neighborhood as follows:

$$\text{Softplus}_\beta(x) = \mathbb{E}_{\epsilon \sim p_\beta}[\text{ReLU}(x - \epsilon)], \tag{6}$$

where $p_\beta(\epsilon) = \frac{\beta}{(e^{\beta\epsilon/2} + e^{-\beta\epsilon/2})^2}$ (Dombrowski et al., 2019). See Appendix F for a short proof. Smilkov et al. (2017); Dombrowski et al. (2019) show that the local average of gradient values can

make the gradient smoother. This means that the gradient obtained with $\mathrm{Softplus}_\beta$ is less noisy and more meaningful than ReLU. We use $\mathrm{Softplus}_\beta$ as an approximation of the activation function ReLU in the pre-trained model. As shown in Fig. 2 (Left), **AAI** first increases then decreases when decreasing $\beta$ in $\mathrm{Softplus}_\beta$ (the smaller the $\beta$, the smoother the classifiers) and **AAI** achieves the largest value at $\beta = 15$. Thus there exits a trade-off between **AAI** and smoothness, and a proper $\beta$ is needed.

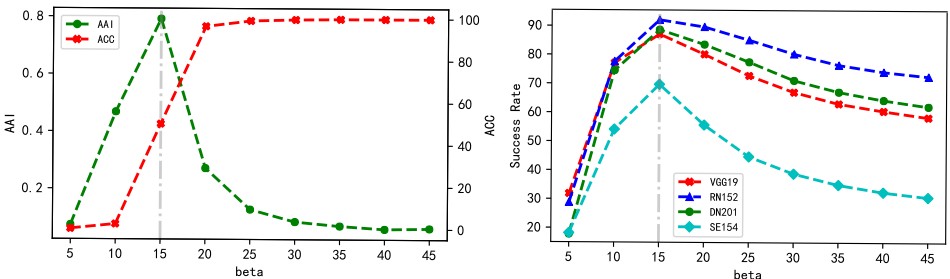

Figure 2: (Left) **AAI** and accuracy of the normal ResNet-50 on ImageNet by replacing ReLU with different $\mathrm{Softplus}_\beta$. (Right) The attack success rate of adversarial examples generated by the pre-trained ResNet-50 modified with different $\mathrm{Softplus}_\beta$. The baseline on each target model is similar to the success rate at $\beta = 45$. See Appendix G for illustration on DenseNet-121.

We conducted an experiment with the PGD attack on the randomly selected 5000 ImageNet validation images which are correctly classified to investigate how the shape-related hyper-parameter $\beta$ affects the adversarial transferability. As shown in Fig. 2 (Right), when the hyper-parameter $\beta$ is too small, $\mathrm{Softplus}_\beta$ cannot approximate ReLU well, which reduces the effectiveness of adversarial perturbation and prediction accuracy. With a proper selection of $\beta = 15$ which is also the maximum of **AAI**, using $\mathrm{Softplus}_\beta$ as an approximation of ReLU can significantly improve the transferability of adversarial examples.

### 3.2.2 UTILIZING DISTRIBUTION-RELEVANT INFORMATION FROM EARLY LAYERS

Maennel et al. (2020) and Boopathy & Fiete (2021) show that the early layers can better capture the local statistics of the inputs than the later layers. This discovery inspires us to observe whether the gradient of the early layers of the model is better aligned with the gradient of the ground truth data distribution. The general ResNet consists of four blocks and the residual module (He et al., 2016) can be expressed as:

$$z_{i+1} = z_i + \mathcal{F}(z_i), \tag{7}$$

where $z_{i+1}$ is the output of residual modules and $z_i$ is the input of residual modules, $\mathcal{F}$ is the residual function. Considering that the information can transfer from shallow layers to deep layers through the skip connection in ResNet-like neural networks, we decrease the weight for certain residual modules to reduce the impact of these modules and enhance the impact of the previous modules from the skip connection. Thus, we modify the residual module as:

$$z_{i+1} = z_i + \lambda_i \cdot \mathcal{F}(z_i), \tag{8}$$

where $0 \leq \lambda_i \leq 1$ can reduce the impact of the certain residual modules. We use the $\lambda_i$ to relatively adjust the influence of different layers.

As shown in Fig. 3, when decreasing $\lambda_{1:4}$ at the same rate on pre-trained ResNet-50 (Paszke et al., 2019), the **AAI** metric first increases then decreases with reaching its maximum at $\lambda_{1:4} = 0.6$. At the same time, different target models obtain the best attack success rate also at $\lambda_{1:4} = 0.6$ (see Fig. 3 (Middle)). This implies that adjusting the influence of different residual branches can help to increase **AAI** as well as transferability. Moreover, we find that decreasing the value of $\lambda_4$ can significantly improve the transferability a lot (from 53% to 71.32%) (Fig. 3 (Right)). This shows that the last block does restrict adversarial transferability and reducing the impact of the last block can greatly improve the transferability. Decreasing $\lambda_{1,2,3}$ for Block1, Block2, and Block3 can slightly improve the adversarial transferability. If we want to achieve the best transferability, we need carefully tuning the hyper-parameters $\lambda_{1:4}$. As our **AAI** metric can be effectively predict the transferability, we use Bayesian optimization to search the best $\lambda_{1:4}$ to maximize **AAI**.

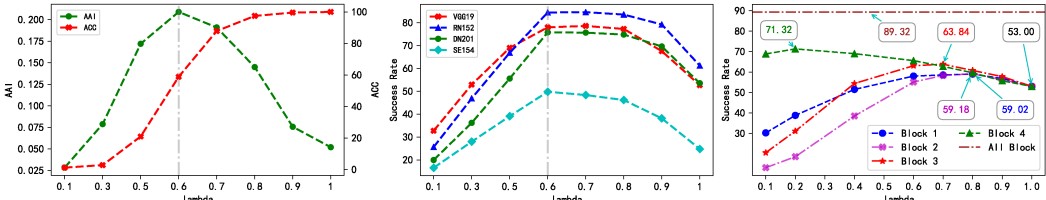

Figure 3: (Left) We illustrate the impact of applying the same $\lambda$ on all the residual modules of the normal ResNet-50 (ImageNet). $\lambda_{1:4} = 0.6$ reaches the maximum of **AAI** while the accuracy of the modified model is only around 60%. (Middle) We illustrate the attack success rate when applying the same $\lambda$ on all the residual modules of the normal ResNet-50. The $\lambda_{1:4} = 0.6$ has the best success rate, which also has the maximum **AAI**. (Right) The impact of applying different $\lambda_i$ on each block. The source model is ResNet-50 and the target model is VGG19. The horizontal line shows the attack success rate when combining the best $\lambda_i$ for each block.

---

**Algorithm 1** Intrinsic Adversarial Attack (**IAA**)

---

**Input:** A pre-trained classifier $f_\theta$; original image $\boldsymbol{x}$; $\ell_\infty$ perturbation radius $\epsilon$; step size $\alpha$; iterations $N$; the classification loss $\ell$;
**Output:** The modified model $f_{\theta,\beta^*,\lambda^*}$, adversarial perturbation $\boldsymbol{\delta}$.

Solve Eq. (9) to find the modified model $f_{\theta,\beta^*,\lambda^*}$, replace ReLU with $\mathrm{Softplus}_{\beta^*}$ and apply $\lambda_i^*$s to residual modules of $f_\theta$.
Initialize $\boldsymbol{\delta} = Uniform(-\epsilon, \epsilon)$.
**for** $i = 1, 2, ..., N$ **do**
    $\boldsymbol{\delta} = \boldsymbol{\delta} + \alpha \cdot sign(\nabla_{\boldsymbol{x}}\ell(f_{\theta,\lambda,\beta}(\boldsymbol{x} + \boldsymbol{\delta}), y))$,
    $\boldsymbol{\delta} = max(min(\boldsymbol{\delta}, \epsilon), -\epsilon)$
**end for**
**return** $\boldsymbol{\delta}$

---

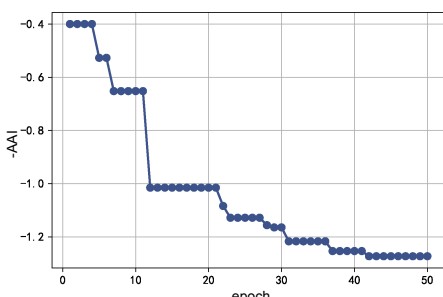

Figure 4: The training loss of Eq. (9) with Bayesian optimization on pre-trained ResNet-50. The search results are $\beta = 20$, $\lambda_1 = 0.98$, $\lambda_2 = 0.87$, $\lambda_3 = 0.73$, $\lambda_4 = 0.19$.

We can also make full use of the distribution-relevant information of a given classifier to increase **AAI** without skip connections. As shown in Appendix J, there are many other ways that can enhance the impact of the early layers to increase **AAI** and can be applied to models without skip connection, but this is not the focus of this paper. Our main contribution is to rethink the adversarial transferability from a novel perspective and reveal that the distribution-relevant information may be the key for boosting adversarial transferability.

### 3.2.3 THE ALGORITHM **IAA**

As analyzed in previous sections, we have verified that the **AAI** is an effective metric for transferability. The smoothness and data distribution information from different layers can be important factors for maximizing **AAI**. In order to find the best parameters, we use Bayesian optimization to optimize

$$\min_{\beta,\lambda} \mathbb{E}_{p_D(\boldsymbol{x},y)}\mathbb{E}_{p(\boldsymbol{v})}\left[\boldsymbol{v}^{\mathrm{T}}\nabla_{\boldsymbol{x}}\frac{\boldsymbol{v}^{\mathrm{T}}\nabla_{\boldsymbol{x}}p_{\theta,\beta,\lambda}(y|\boldsymbol{x})}{\|\nabla_{\boldsymbol{x}}p_{\theta,\beta,\lambda}(y|\boldsymbol{x})\|_2}\right]. \tag{9}$$

We propose **Intrinsic Adversarial Attack (IAA)** as shown in Alg. 1, which replaces the activation function ReLU with $\mathrm{Softplus}_\beta$ and applies corresponding decay parameters $\lambda_i$ searched by Eq. (9) to different residual modules. Fig. 4 shows the training curve on pre-trained ResNet-50, see Appendix C for setting details. The experiments are shown in next section. We analyze the computation cost in Appendix M and show other optimization methods to find the proper structural parameters in Appendix O.

Table 1: Transferability against normal models: the success rates of black-box attacks (untargeted) crafted on RN50, RN152, DN121 and DN201.

| Source | Attack | VGG19 | RN50 | RN152 | DN121 | DN201 | SE154 | IncV3 | IncV4 | IncRes |
|---|---|---|---|---|---|---|---|---|---|---|
| RN50 | PGD | 53.00% | **99.94%** | 61.26% | 55.62% | 53.56% | 24.78% | 20.86% | 21.96% | 17.60% |
| | MI | 64.86% | 99.72% | 73.22% | 73.50% | 64.33% | 47.20% | 39.08% | 37.35% | 25.26% |
| | DI | 75.06% | 99.78% | 81.65% | 81.98% | 74.80% | 52.42% | 42.58% | 39.30% | 27.12% |
| | ILA | 83.56% | 99.75% | 92.46% | 88.40% | 85.24% | 61.44% | 49.94% | 48.34% | 35.74% |
| | SGM | 82.72% | 99.82% | 88.40% | 83.56% | 80.34% | 61.30% | 53.72% | 49.83% | 42.86% |
| | IR | 82.46% | 99.80% | 85.24% | 84.35% | 82.10% | 64.20% | 54.60% | 51.05% | 46.78% |
| | **IAA** | **96.40%** | 99.88% | **98.32%** | **97.46%** | **96.64%** | **84.74%** | **76.28%** | **71.38%** | **64.32%** |
| RN152 | PGD | 49.32% | 72.72% | **99.91%** | 53.44% | 51.00% | 26.32% | 23.50% | 22.58% | 18.72% |
| | MI | 65.42% | 83.40% | 99.82% | 77.60% | 75.79% | 53.00% | 46.50% | 43.32% | 33.08% |
| | DI | 74.01% | 88.18% | 99.78% | 79.46% | 77.81% | 57.49% | 50.28% | 47.16% | 35.10% |
| | ILA | 66.20% | 90.44% | 99.88% | 75.48% | 73.80% | 50.32% | 42.32% | 41.30% | 29.98% |
| | SGM | 80.40% | 96.10% | 99.87% | 85.80% | 82.76% | 61.90% | 53.16% | 49.24% | 43.30% |
| | IR | 73.20% | 92.70% | 99.84% | 83.43% | 80.60% | 64.00% | 53.60% | 50.30% | 48.00% |
| | **IAA** | **94.46%** | **99.32%** | 99.87% | **96.58%** | **95.06%** | **82.46%** | **76.34%** | **71.04%** | **58.34%** |
| DN121 | PGD | 56.78% | 63.22% | 52.76% | **99.94%** | 71.98% | 31.46% | 24.92% | 26.82% | 20.64% |
| | MI | 68.36% | 74.18% | 72.88% | 99.80% | 89.56% | 58.58% | 52.22% | 45.35% | 35.24% |
| | DI | 73.68% | 79.56% | 74.72% | 99.85% | 89.40% | 53.34% | 53.65% | 47.94% | 37.72% |
| | ILA | 87.76% | 90.38% | 83.42% | 99.82% | 95.32% | 65.02% | 58.64% | 57.36% | 40.76% |
| | SGM | 80.18% | 88.54% | 80.54% | 99.79% | 92.70% | 64.92% | 54.62% | 49.82% | 37.76% |
| | IR | 82.56% | 86.14% | 85.20% | 99.82% | 95.30% | 72.20% | 62.22% | 62.10% | 56.00% |
| | **IAA** | **96.80%** | **96.78%** | **93.90%** | 99.88% | **97.70%** | **88.34%** | **88.24%** | **87.36%** | **77.18%** |
| DN201 | PGD | 57.76% | 70.68% | 59.08% | 83.06% | **99.89%** | 40.60% | 33.80% | 32.46% | 23.80% |
| | MI | 75.09% | 82.46% | 76.39% | 88.18% | 99.84% | 64.38% | 59.62% | 54.85% | 39.40% |
| | DI | 78.11% | 85.34% | 78.18% | 90.20% | 99.81% | 61.75% | 60.04% | 56.15% | 40.56% |
| | ILA | 88.56% | 94.78% | 90.02% | 98.02% | 99.72% | 76.34% | 67.78% | 65.36% | 49.50% |
| | SGM | 82.72% | 91.72% | 86.60% | 96.40% | 99.67% | 72.20% | 62.34% | 56.36% | 45.42% |
| | IR | 76.74% | 90.46% | 85.40% | 95.39% | 99.74% | 73.60% | 59.80% | 63.00% | 56.60% |
| | **IAA** | **96.32%** | **96.98%** | **93.82%** | **98.10%** | 99.78% | **87.98%** | **88.26%** | **87.02%** | **79.12%** |

# 4 EXPERIMENTS

## 4.1 IMPLEMENTATION

**Attack Setting.** We use a black-box threat model to test the adversarial transferability. Adversarial examples are generated by attacking the source models, and then these examples are applied to attack target models. The architecture of the source model is different from that of the target model. In terms of attack strength, we follow the same standard setting for all attack methods (Xie et al., 2019; Wu et al., 2019). We constrain the adversarial perturbation within the $\ell_\infty$ ball of radius $\epsilon = 16/255$ with respect to pixel value in $[0, 1]$ and set the step size $\alpha$ to $2/255$. The iteration steps in all the experiments are set to $10$. All experiments in this paper are run on Tesla V100. The experiments are repeated 5 times with different random seeds and we just show the maximum standard deviations in the caption of tables due to page limitations. See Appendix C for more attack setting.

**Target Models and Source Models.** We conduct experiments on both normal target models and robust target models. For normal target models, we choose 9 models: VGG19 (Simonyan & Zisserman, 2015), ResNet-50 (RN50), ResNet-152 (RN152) (He et al., 2016), DenseNet-121 (DN121), DenseNet-201 (DN201) (Huang et al., 2017), 154 layers Squeeze-and-Excitation network (SE154) (Hu et al., 2018), Inception V3 (IncV3) (Szegedy et al., 2016), Inception V4 (IncV4), and Inception-ResNet V2 (IncRes) (Szegedy et al., 2017), and we use the pre-trained models in PyTorch (Paszke et al., 2019). For robust target models, we consider 3 robustly trained models using ensemble adversarial training (Tramèr et al., 2018): IncV3ens3 (ensemble of 3 IncV3 models), IncV3ens4 (ensemble of 4 IncV3 models) and IncResV2ens3 (ensemble of 3 IncResV2 models). We choose 4 models as source models: ResNet-50, ResNet-152, DenseNet-121, DenseNet-201. We also choose VGG16, VGG19 (models without skip connection) as source model in Appendix J.

## 4.2 TRANSFERABILITY AGAINST NORMAL MODELS

Firstly, we focus on **untargeted** adversarial attacks. We compare different attack methods on four source models against nine normal target models. In all transfer scenarios, **IAA** outperforms existing methods by a large margin consistently against different target models. The results are reported in

Tab. 1. For transfer ResNet-50 → VGG19, **IAA** achieves a success rate of 96.40% which is 43.40% and 13.68% higher than PGD and SGM respectively. For transfer ResNet-50 → Inception-ResNet V2, IR performs the best in the existing methods (attack success rate 46.78%) while our **IAA** can achieve a new state-of-the-art success rate 64.32%.

In Tab. 1, we show that **IAA** surpasses the other existing methods when attacking against normal models under Top-1 accuracy. Top-5 accuracy, which checks if the ground truth label is among the five predictions with the highest probability, is also an important metric. In Tab. 2, we show the first five predictions of the target model. For transfer ResNet-50 → Inception V3, the top-5 accuracy on adversarial examples crafted by PGD is 96.66%. SGM is an efficient method to enhance adversarial transferability, but the top-5 accuracy is as high as 84.28% for transfer ResNet-50 → Inception V3. Our **IAA** drives the examples to the low-density region by matching the adversarial attack and intrinsic attack. Thus the target models can hardly give a proper prediction, and the **top-5 accuracy is significantly lower** than other methods.

Table 2: Top-5 accuracy on different target models when predicting the black-box adversarial examples crafted (untargeted) on RN50, RN152, DN121 and DN201. The best results are in bold.

| | Attack | VGG19 | DN121 | DN201 | SE154 | IncV3 | | Attack | VGG19 | RN50 | RN152 | SE154 | IncV3 |
|---|---|---|---|---|---|---|---|---|---|---|---|---|---|
| RN50 | PGD | 83.52 | 84.16 | 86.92 | 93.00 | 96.66 | DN121 | PGD | 83.86 | 82.84 | 90.70 | 95.00 | 95.78 |
| | SGM | 58.68 | 63.52 | 68.46 | 75.62 | 84.28 | | SGM | 61.16 | 55.36 | 67.18 | 71.46 | 81.24 |
| | **IAA** | **5.67** | **4.37** | **6.60** | **18.68** | **29.36** | | **IAA** | **7.14** | **7.82** | **14.04** | **22.50** | **24.80** |
| RN152 | PGD | 89.68 | 87.70 | 89.94 | 95.18 | 96.60 | DN201 | PGD | 79.46 | 73.36 | 81.62 | 87.92 | 92.94 |
| | SGM | 62.30 | 61.62 | 67.4 | 74.60 | 82.00 | | SGM | 60.76 | 52.46 | 61.46 | 68.64 | 78.78 |
| | **IAA** | **8.46** | **6.32** | **9.92** | **25.42** | **33.4** | | **IAA** | **10.08** | **7.10** | **15.40** | **24.48** | **23.82** |

Table 3: Transferability against normal models: the success rates (%) of black-box attacks crafted (targeted) on RN50, RN152, DN121 and DN201. The best results are in bold. The results are averaged on 8 different target classes.

| | Attack | VGG19 | RN50 | RN152 | DN121 | DN201 | | Attack | VGG19 | RN50 | RN152 | DN121 | DN201 |
|---|---|---|---|---|---|---|---|---|---|---|---|---|---|
| RN50 | PGD | 0.40 | 98.96 | 1.18 | 0.78 | 0.4 | DN121 | PGD | 0.12 | 0.15 | 0.10 | 99.64 | 0.14 |
| | SGM | 4.48 | 99.98 | 6.92 | 6.50 | 3.14 | | SGM | 0.55 | 1.26 | 0.75 | 99.73 | 1.03 |
| | **IAA** | **33.68** | **100** | **53.72** | **52.46** | **38.02** | | **IAA** | **8.80** | **14.16** | **6.58** | **99.92** | **16.74** |
| RN152 | PGD | 0.24 | 2.78 | 99.76 | 0.74 | 0.38 | DN201 | PGD | 0.15 | 0.10 | 0.12 | 0.57 | 99.75 |
| | SGM | 1.62 | 5.14 | **100** | 4.28 | 1.08 | | SGM | 0.63 | 1.84 | 1.14 | 3.68 | 99.84 |
| | **IAA** | **15.04** | **48.18** | 99.90 | **26.70** | **14.82** | | **IAA** | **16.34** | **27.64** | **17.72** | **31.94** | **99.84** |

Secondly, we focus on **targeted** attacks. Liu et al. (2017); Yang et al. (2021b) show that generating transferable targeted adversarial examples is much more difficult than generating transferable untargeted adversarial examples. As we analyzed before, when crafting the untargeted attack, the intrinsic adversarial attack can drive adversarial examples to low-density regions. Similarly, when crafting the targeted attack, **IAA** can move the adversarial images towards the high-density region of the target class along the direction of $\nabla_{\boldsymbol{x}} \log p_\theta(\boldsymbol{x}, y_{target})$. Thus **IAA** can also improve the adversarial transferability of targeted examples, and we show the results in Tab. 3. For transfer ResNet-50 → VGG19 (targeted), **IAA** achieves a success rate of 33.68% which is 33.28% and 29.20% higher than PGD and SGM respectively.

We also evaluate the transferability of **one-step** attacks in Fig. 5. SGM can greatly enhance the transferability of the multi-step attack, but it can only slightly improve the effect of the one-step attack. DI performs better in the transferability of the one-step attack than SGM. We think that diversity inputs may help to reduce the noise in gradient and get better adversarial perturbation. Our **IAA** can significantly improve the adversarial transferability of the one-step attack. This is because that the perturbation crafted by **IAA** is aligned with the intrinsic attack $-\nabla_{\boldsymbol{x}} \log p_D(\boldsymbol{x}, y)$ and can efficiently drive the examples towards the low-density region. It is difficult for the target models to predict accurately for examples in the low-density region.

## 4.3 TRANSFERABILITY AGAINST ROBUSTLY TRAINED MODELS

The success rates of our **IAA** and other methods against the robust target models (using ensemble adversarial training) are reported in Tab. 4. TI was originally proposed for the robust models. Compared with the existing methods, **IAA** has a large improvement in all scenarios. For transfer ResNet-50 → $IncV3_{ens3}$, **IAA** achieves a success rate of 38.72% which is 31.2% and 5.91% higher than PGD and SGM respectively. We think that both the robust model and the normal model are trained on the same dataset. **IAA** perturbs the distribution-relevant information, which leads to

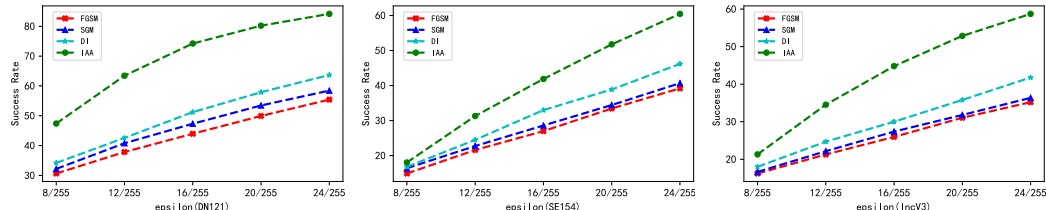

Figure 5: The attack success rate (one-step attack) of different methods against different target models. The source model is RN50. The horizontal axis represents different attack strengths. (Left) The target model is DN121. (Middle) The target model is SE154. (Right) The target model is IncV3.

misclassification of the robust models. We show that IAA can also effectively penetrate many other defense methods in Appendix N.

Table 4: Transferability against robustly trained models: the success rates of black-box attacks crafted on source models: ResNet-50, ResNet-152, DenseNet121 and DenseNet201.

| Source | Attack | $IncV3_{ens3}$ | $IncV3_{ens4}$ | $IncRes_{ens3}$ | Source | Attack | $IncV3_{ens3}$ | $IncV3_{ens4}$ | $IncRes_{ens3}$ |
|--------|--------|--------|--------|--------|--------|--------|--------|--------|--------|
|        | PGD | 7.52% | 9.50% | 5.58% |        | PGD | 13.80% | 13.10% | 7.90% |
|        | TI | 30.06% | 23.48% | 16.20% |        | TI | 28.10% | 27.50% | 21.20% |
| RN50 | SGM | 32.81% | 26.80% | 20.02% | DN121 | SGM | 40.20% | 36.83% | 27.86% |
|        | IR | 29.56% | 22.18% | 13.90% |        | IR | 26.00% | 24.10% | 15.90% |
|        | **IAA** | **38.72%** | **33.62%** | **24.32%** |        | **IAA** | **59.52%** | **43.38%** | **43.16%** |
|        | PGD | 12.20% | 10.80% | 5.70% |        | PGD | 18.16% | 15.30% | 10.40% |
|        | TI | 35.97% | 32.81% | 20.16% |        | TI | 42.76% | 42.01% | 34.28% |
| RN152 | SGM | 31.57% | 27.77% | 20.84% | DN201 | SGM | 41.45% | 37.85% | 29.41% |
|        | IR | 29.34% | 24.33% | 13.76% |        | IR | 38.90% | 38.45% | 28.70% |
|        | **IAA** | **43.28%** | **37.88%** | **26.78%** |        | **IAA** | **61.02%** | **53.80%** | **46.34%** |

### 4.4 ENSEMBLE-BASED ATTACKS

Wu et al. (2019); Wang et al. (2020) show that the ensemble-based strategy (Liu et al., 2017) can improve the performance of their method. This section, shows the results in Tab. 5, and the attacks are crafted on an ensemble of ResNet-34, ResNet-152, and DenseNet-201. We show that our **IAA** with the ensemble-based strategy can generate more transferable perturbation than other methods. For transfer Ensemble model → IncV4, **IAA** achieves a success rate of 90.70% which is 7.73% and 10.63% higher than SGM and IR respectively.

Table 5: Transferability against different models: the success rates of black-box attacks (untargeted) crafted on an ensemble of 3 models (RN34, RN152 and DN201). The best results are in bold.

| Model | Attack | VGG16 | VGG19 | SE154 | IncV3 | IncV4 | IncRes |
|-------|--------|-------|-------|-------|-------|-------|--------|
|          | PGD | 86.60% | 86.69% | 69.65% | 69.95% | 59.30% | 53.91% |
|          | TI | 85.84% | 84.35% | 71.67% | 67.22% | 66.02% | 56.83% |
| Ensemble | DI | 96.85% | 96.34% | 89.72% | 87.53% | 85.04% | 81.11% |
|          | SGM | 96.84% | 97.36% | 90.40% | 87.86% | 82.97% | 80.93% |
|          | IR | 91.50% | 91.16% | 86.12% | 81.69% | 80.07% | 79.34% |
|          | **IAA** | **99.60%** | **99.68%** | **95.34%** | **93.60%** | **90.70%** | **81.54%** |

## 5 CONCLUSION

In this paper, we rethink the adversarial perturbation from the distribution perspective and show that the alignment between the adversarial attack and intrinsic attack (**AAI**) is an effective metric for predicting the adversarial transferability. Furthermore, we propose a novel algorithm (**IAA**) which maximizes **AAI** for generating adversarial examples in the low-density region by Bayesian optimization. We have conducted experiments on four source models, nine normal target models (including untargeted attack and targeted attack) and three robust target models and show that our methods surpass other existing methods by a large margin. Our findings can motivate new research into the adversarial transferability and adversarial examples from the perspective of data distribution and open up further challenges for the defense against black-box attacks. What's more, **IAA** can also fool the image search engines (Google Reverse Image Search and Baidu Reverse Image Search) as shown in Appendix A.

ETHICS STATEMENT

In this paper, we propose a novel method **IAA** to boost adversarial transferability. IAA can craft stronger adversarial examples from the source model to attack unknown target models than other existing methods. Our findings can motivate new research into the adversarial transferability and adversarial examples from the data distribution perspective and open up further challenges for the defense against black-box attacks. Our goal is to understand current deep learning's weaknesses and make deep models more robust and transparent. We did not use crowdsourcing and did not conduct research with human subjects in our experiments. We cited the creators when using existing assets (e.g., code, data, models).

REPRODUCIBILITY STATEMENT

We show the proofs of Theorem 1 in Appendix E. Our **IAA** is an appealingly simple method. We specify the settings of hyper-parameters and how they were chosen in Appendix C. We repeat experiments 5 times with different random seeds and show the standard deviation in the caption of tables. We plan to open the source code to reproduce the main experimental results later.

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

## A  FOOLING THE IMAGE SEARCH ENGINES

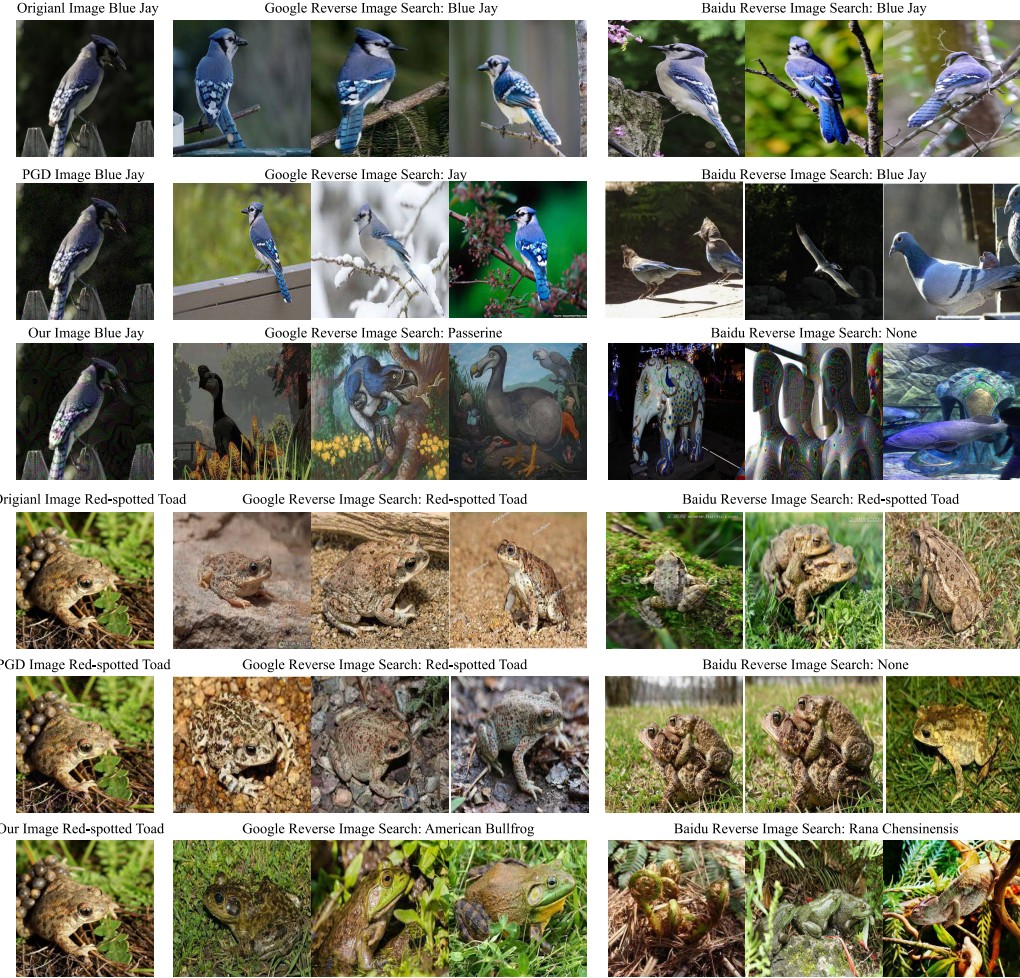

Figure 6: We show that the adversarial perturbation generated by our **IAA** is imperceptible to human observers but can fool the images search engines. The first three lines show the recognition results of the image Blue Jay in different image search engines. Baidu Reverse Images Search can't give the classification results when identifying our adversarial examples, and the similar pictures searched are not matched with Blue Jay. The last three lines show the recognition results of Red-spotted Toad in the image search engines. Both Google Reverse Images Search and Baidu Reverse Images Search give wrong recognition results.

The previous experiment mainly verified the attack effect of **IAA** on the pre-trained classification model. In this section we show that **IAA** can fool the image search engines successfully. As shown in Fig. 6, the original image can be correctly identified by both **Google Reverse Images Search** and **Baidu Reverse Images Search**. However, the adversarial images generated by our **IAA** can fool the image search engines.

## B  HIGH TRANSFERABLE EXAMPLES

We use a source model RN50 to generate adversarial examples with widely used PGD attack. If the original image can be correctly classified by different targeted models (VGG19, RN50, RN152, DN121, DN201 and SE154) and its adversarial counterpart can fool all the target models, we save this original image. Then, we randomly save 1000 samples from ImageNet evaluation datasets. We named these samples Low-Density Data (L_Data). If the original images can be correctly classified

by different targeted models (VGG19, RN50, RN152, DN121, DN201 and SE154) and their adversarial counterparts can only fool the source models, we save this original image. Then, we randomly save 1000 samples from ImageNet evaluation datasets. We named these samples High-Density Data (H_Data). We find an intriguing phenomena: LDD with random noise can fool different target models, while the attack success rate is much lower when applying the same strength of random noise to HDD. Specifically when we injecting Gaussian noise ($\epsilon$=16/255) in LDD, 23.3% of the samples can fool the target model VGG19. However, when we injecting Gaussian noise ($\epsilon$=16/255) in HDD, only 0.7% of the samples can fool the target model VGG19. What's more, LDD can be correctly classified by different models but their adversarial counterparts generated by different source models have strong transferability against different target models (as shown in Fig. 3.1 and Fig. 7). In Fig. 8, we show some samples of LDD.

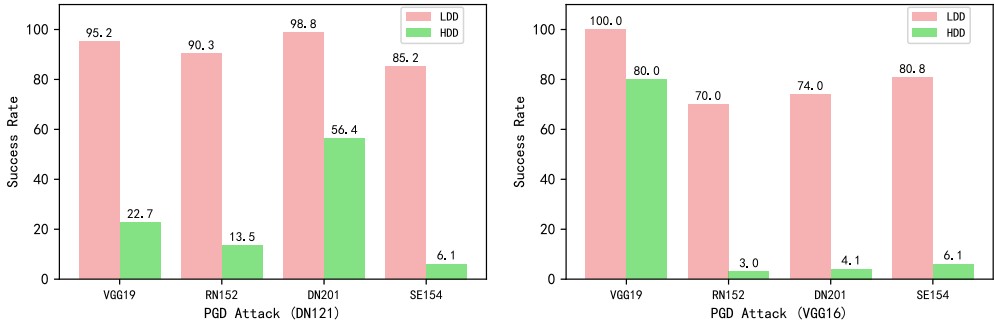

Figure 7: The histogram shows the attack success rate of the adversarial examples for LDD and HDD using PGD ($\ell_\infty$, $\epsilon$=16/255) against different target models (VGG19, RN152, DN201 and SE154). The source model is DenseNet-121 (Left) and VGG16 (Right).

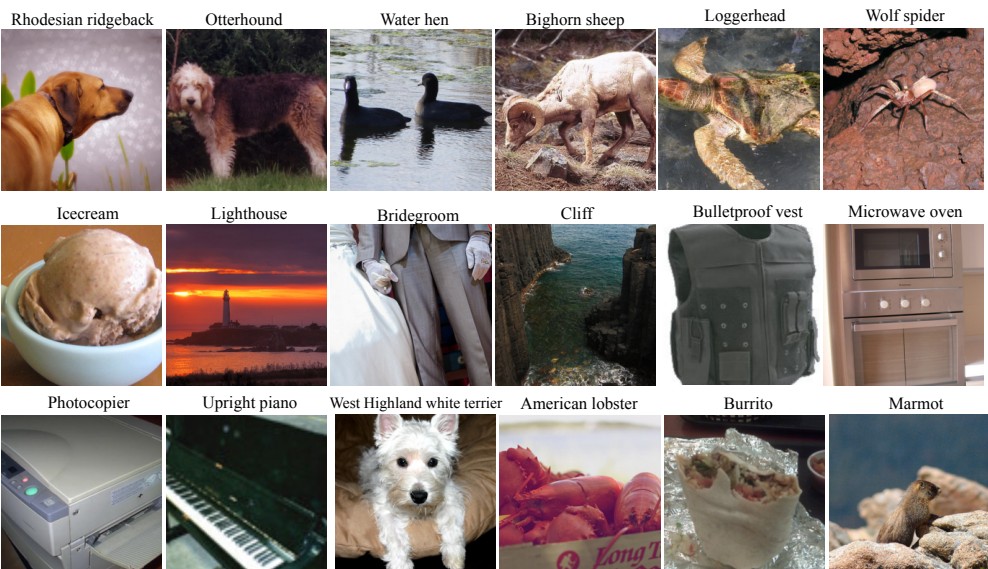

Figure 8: Some samples which have high adversarial transferable counterparts. These samples can be correctly classified by different target models.

## C   ATTACK SETTING

We choose the baseline attack PGD (Madry et al., 2018) and 5 state-of-the-art transfer attacks: MI attack (Dong et al., 2018), DI attack (Xie et al., 2019), ILA attack (Huang et al., 2019), SGM attack (Wu et al., 2019) and IR attack (Wang et al., 2020). And we take TI attack (Dong et al., 2019) which

is proposed to attack robust model into consideration when attacking against robust models. All the source models are trained on ImageNet training set.

The **scikit-optimize** [1] is a simple and efficient library to minimize black-box functions, which helps us to search the best structure hyper-parameters for a pre-trained classifier to minimize the Eq. (9). We use Batesian optimization to optimize the Eq. (9) to search the hyper-parameters $\beta$ and $\lambda_i$. For ResNet-50, the search results are $\beta = 20$, and $\lambda_1$ applied to the residual modules in Block1 is 0.98, $\lambda_2$ applied to the residual modules in Block1 is 0.87, $\lambda_3$ applied to the residual modules in Block3 is 0.73, $\lambda_4$ applied to the residual modules in Block4 is 0.19. For ResNet-152, the search results are $\beta = 32$, and $\lambda_1$ applied to the residual modules in Block1 is 0.89, $\lambda_2$ applied to the residual modules in Block1 is 0.88, $\lambda_3$ applied to the residual modules in Block3 is 0.70, $\lambda_4$ applied to the residual modules in Block4 is 0.20. For DenseNet-121 and DenseNet-201, the search results are $\beta = 35$, and $\lambda_{1,2,3}$ applied to the residual modules in Block1, Block2 and Block3 is 0.80, $\lambda_4$ applied to the residual modules in Block4 is 0.44.

## D    SIMILARITY BETWEEN ADVERSARIAL PERTURBATION

We illustrate frequency histogram of cosine similarity between adversarial perturbation generated by different models in Fig. 9. We also illustrate frequency histogram of Pearson Correlation Coefficient (PCC) between adversarial perturbation generated by different models in Fig. 10. Perturbation generated by different models using **IAA** have stronger correlation than other methods (PGD and SGM), which means that **IAA** reduces the dependence on models and generate more distribution relevant perturbation.

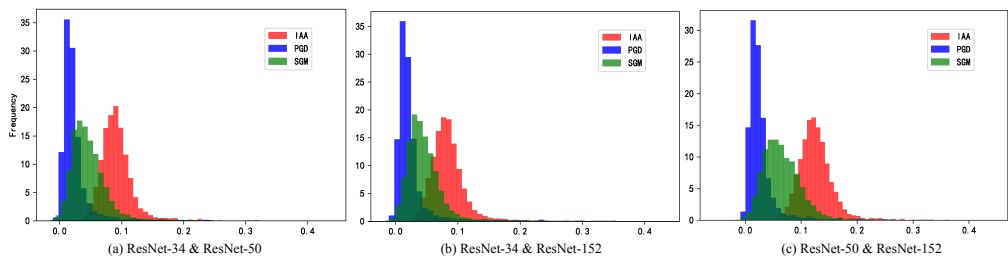

Figure 9: Frequency histogram of cosine similarity between adversarial perturbations generated by ResNet-34, ResNet-50 and ResNet-152. The perturbations generated by our **IAA** methods are closer to each other with different source model than perturbations generated by PGD and SGM.

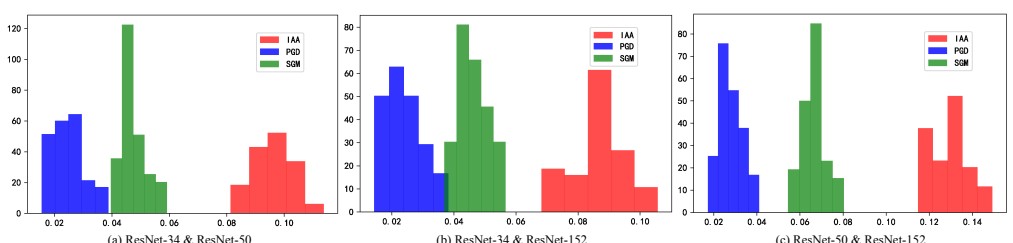

Figure 10: Frequency histogram of Pearson Correlation Coefficient between adversarial perturbations generated by ResNet-34, ResNet-50 and ResNet-152. The perturbations generated by our **IAA** methods are closer to each other with different source model than perturbations generated by PGD and SGM.

---

[1]The website of scikit-optimize is https://scikit-optimize.github.io/stable/

# E PROOF: THEOREM 1

The proof is similar as Hyvärinen & Dayan (2005) and Song et al. (2020).

$$
\begin{aligned}
\mathbf{AAI} &\triangleq \mathbb{E}_{p_D(\boldsymbol{x},y)}\left[\frac{\nabla_{\boldsymbol{x}}\log p_{\theta,\Lambda}(y|\boldsymbol{x})}{\|\nabla_{\boldsymbol{x}}\log p_{\theta,\Lambda}(y|\boldsymbol{x})\|_2}\cdot\nabla_{\boldsymbol{x}}\log p_D(\boldsymbol{x},y)\right] \\
&\overset{(I)}{=} \int dy\int d\boldsymbol{x}\left[\frac{\nabla_{\boldsymbol{x}}p_{\theta,\Lambda}(y|\boldsymbol{x})}{\|\nabla_{\boldsymbol{x}}p_{\theta,\Lambda}(y|\boldsymbol{x})\|_2}\cdot\nabla_{\boldsymbol{x}}\log p_D(\boldsymbol{x},y)\right]p_D(\boldsymbol{x},y) \\
&= \int dy\int d\boldsymbol{x}\left[\frac{\nabla_{\boldsymbol{x}}p_{\theta,\Lambda}(y|\boldsymbol{x})}{\|\nabla_{\boldsymbol{x}}p_{\theta,\Lambda}(y|\boldsymbol{x})\|_2}\cdot\nabla_{\boldsymbol{x}}p_D(\boldsymbol{x},y)\right] \\
&\overset{(II)}{=} \int dy\left\{\lim_{R\to\infty}\int dS_R\frac{\boldsymbol{n}\cdot\nabla_{\boldsymbol{x}}p_{\theta,\Lambda}(y|\boldsymbol{x})}{\|\nabla_{\boldsymbol{x}}p_{\theta,\Lambda}(y|\boldsymbol{x})\|_2}p_D(\boldsymbol{x},y)-\int d\boldsymbol{x}\left[\nabla_{\boldsymbol{x}}\cdot\frac{\nabla_{\boldsymbol{x}}p_{\theta,\Lambda}(y|\boldsymbol{x})}{\|\nabla_{\boldsymbol{x}}p_{\theta,\Lambda}(y|\boldsymbol{x})\|_2}\right]p_D(\boldsymbol{x},y)\right\} \\
&= \int dy\left\{\lim_{R\to\infty}\int dS_R\frac{\boldsymbol{n}\cdot\nabla_{\boldsymbol{x}}p_{\theta,\Lambda}(y|\boldsymbol{x})}{\|\nabla_{\boldsymbol{x}}p_{\theta,\Lambda}(y|\boldsymbol{x})\|_2}p_D(\boldsymbol{x},y)\right\}-\mathbb{E}_{p_D(\boldsymbol{x},y)}\left[\nabla_{\boldsymbol{x}}\cdot\frac{\nabla_{\boldsymbol{x}}p_{\theta,\Lambda}(y|\boldsymbol{x})}{\|\nabla_{\boldsymbol{x}}p_{\theta,\Lambda}(y|\boldsymbol{x})\|_2}\right] \\
&\overset{(III)}{=} -\mathbb{E}_{p_D(\boldsymbol{x},y)}\left[\nabla_{\boldsymbol{x}}\cdot\frac{\nabla_{\boldsymbol{x}}p_{\theta,\Lambda}(y|\boldsymbol{x})}{\|\nabla_{\boldsymbol{x}}p_{\theta,\Lambda}(y|\boldsymbol{x})\|_2}\right].
\end{aligned}
\tag{10}
$$

The log function can be moved for the equality (I) because the normalized $\nabla_x p(y|x)$ and $\nabla_x \log p(y|x)$ are the same direction, as:

$$
\frac{\nabla_x\log p(y|x)}{\|\nabla_x\log p(y|x)\|_2}=\frac{p(y|x)^{-1}\nabla_x p(y|x)}{\|p(y|x)^{-1}\nabla_x p(y|x)\|_2}=\frac{p(y|x)^{-1}\nabla_x p(y|x)}{p(y|x)^{-1}\|\nabla_x p(y|x)\|_2}=\frac{\nabla_x p(y|x)}{\|\nabla_x p(y|x)\|_2},\tag{11}
$$

where we use the formula: $\nabla_x\log f(x)=f(x)^{-1}\nabla_x f(x)$.

Take 1d case for example, we use integration by part formula for the equality (II):

$$
\int_{-\infty}^{+\infty}f^{'}(x)g^{'}(x)dx=\lim_{R\to\infty}f^{'}(x)g(x)|_{-R}^{+R}-\int_{-\infty}^{+\infty}f^{''}(x)g(x)dx,\tag{12}
$$

The equality (III) holds by Holder Inequality (Bartle, 2014):

$$
\left|\int dS_R\frac{\boldsymbol{n}\cdot\nabla_{\boldsymbol{x}}p_{\theta,\Lambda}(y|\boldsymbol{x})}{\|\nabla_{\boldsymbol{x}}p_{\theta,\Lambda}(y|\boldsymbol{x})\|_2}p_D(\boldsymbol{x},y)\right|\leq\left[\int p_D^2(\boldsymbol{x},y)dS_R\right]^{\frac{1}{2}}\to 0,\ R\to\infty.
$$

For a distribution of a random vector v such that $\mathbb{E}_{p(\boldsymbol{v})}[\boldsymbol{v}\boldsymbol{v}^{\mathrm{T}}]=I$, then we have

$$
\begin{aligned}
\mathbf{AAI} &= -\mathbb{E}_{p_D(\boldsymbol{x},y)}\left[\nabla_{\boldsymbol{x}}\cdot\frac{\nabla_{\boldsymbol{x}}p_{\theta,\Lambda}(y|\boldsymbol{x})}{\|\nabla_{\boldsymbol{x}}p_{\theta,\Lambda}(y|\boldsymbol{x})\|_2}\right] \\
&= -\mathbb{E}_{p_D(\boldsymbol{x},y)}\mathbb{E}_{p(\boldsymbol{v})}\left[\nabla_{\boldsymbol{x}}\cdot\frac{\boldsymbol{v}\boldsymbol{v}^{\mathrm{T}}\nabla_{\boldsymbol{x}}p_{\theta,\Lambda}(y|\boldsymbol{x})}{\|\nabla_{\boldsymbol{x}}p_{\theta,\Lambda}(y|\boldsymbol{x})\|_2}\right] \\
&= -\mathbb{E}_{p_D(\boldsymbol{x},y)}\mathbb{E}_{p(\boldsymbol{v})}\left[\boldsymbol{v}^{\mathrm{T}}\nabla_{\boldsymbol{x}}\frac{\boldsymbol{v}^{\mathrm{T}}\nabla_{\boldsymbol{x}}p_{\theta,\Lambda}(y|\boldsymbol{x})}{\|\nabla_{\boldsymbol{x}}p_{\theta,\Lambda}(y|\boldsymbol{x})\|_2}\right].
\end{aligned}
\tag{13}
$$

We use the property that random vector $v$ is independent of $(x,y)$ in Eq. 13, thus we can switch $\nabla_x$ and $v$.

# F PROOF: SOFTPLUS IS THE EXPECTATION OF RELU

$$
\mathrm{Softplus}_\beta(x)=\frac{1}{\beta}log(1+\exp(\beta x)),\tag{14}
$$

where $\beta$ is the shape-related hyper-parameter. The derivative of this function is

$$
\frac{d}{dx}\mathrm{Softplus}_\beta(x)=\frac{1}{1+\exp(-\beta x)}.\tag{15}
$$

When the $p_\beta(\epsilon)$ is implicitly defined, we express $\text{Softplus}_\beta(x)$ as:

$$\text{Softplus}_\beta(x) = \int_{-\infty}^{+\infty} p_\beta(\epsilon)\text{ReLU}(x - \epsilon)d\epsilon. \tag{16}$$

With respect to x, the differential of the equation is:

$$\frac{d}{dx}\text{Softplus}_\beta(x) = \int_{-\infty}^{+\infty} p_\beta(\epsilon)\Theta(x - \epsilon)d\epsilon = \int_{-\infty}^{x} p_\beta(\epsilon)d\epsilon, \tag{17}$$

where $\Theta(x) = I(x > 0)$ is the Heaviside step function. From Eq. (15), the differential of Eq. (17) is:

$$\frac{\beta}{(e^{\beta\epsilon/2} + e^{-\beta\epsilon/2})^2} = p_\beta(\epsilon). \tag{18}$$

Thus, the $\text{Softplus}_\beta$ can be expressed as the expectation of ReLU in a neighborhood as follows:

$$\text{Softplus}_\beta(x) = \mathbb{E}_{\epsilon \sim p_\beta}[\text{ReLU}(x - \epsilon)], \tag{19}$$

where $p_\beta(\epsilon) = \frac{\beta}{(e^{\beta\epsilon/2} + e^{-\beta\epsilon/2})^2}$.

## G  THE INFLUENCE OF STRUCTURE PARAMETERS ON DENSENET-121

We first show the influence of the shape parameter $\beta$ when we use replace ReLU with $Softplus_\beta$ in Fig. 11 (Left). A proper $\beta$ can improve the adversarial transferability. The general ResNet and DenseNet consists of four blocks. We illustrate the influence of applying $\lambda_i$ on different blocks of ResNet-50 in paper. Here, we illustrate the influence of applying $\lambda_i$ on different blocks of DenseNet-121 in Fig. 11 (Right). The target model is VGG19. $\lambda_i$ applied to Block4 has great influence on adversarial transferability, and $\lambda_i$=0.4 can improve the attack success rate from $56.78\%$ to $67.70\%$. $\lambda_i$ applied to Block1, Block2, and Block3 can slightly improve the adversarial transferability. When we apply the best $\lambda_i$ to each block, the attack success rate can reach $85.08\%$, which has exceeded state-of-the-art methods SGM ($80.18\%$) and IR ($82.56\%$).

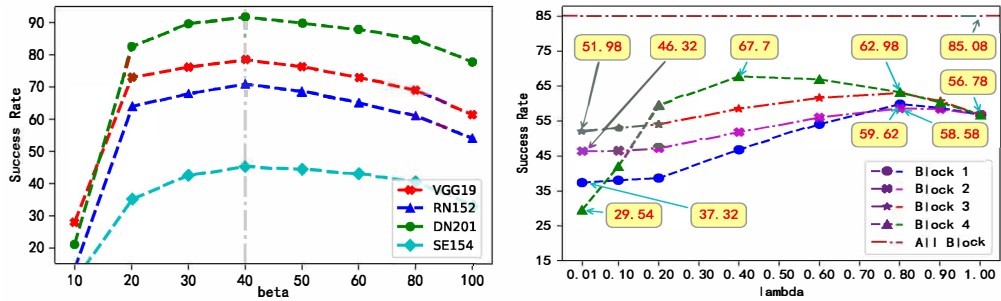

Figure 11: (Left) The attack success rate of adversarial examples generated by the pre-trained (DenseNet-121) modified with different $Softplus_\beta$. (Right) The influence of applying $\lambda$ on different blocks. The source model is DenseNet-121 and the target model is VGG19. The green line with triangle mark shows attack success rate when applying $\lambda$ to Block4 (the last block), and $\lambda = 0.4$ has the best effect. The red line with star mark shows attack success rate when applying $\lambda$ to Block3, and $\lambda = 0.8$ has the best effect. The purple line with cross mark shows attack success rate when applying $\lambda$ to Block2, and $\lambda = 0.8$ has the best effect. The blue line with circular mark shows attack success rate when applying $\lambda$ to Block1, and $\lambda = 0.8$ has the best effect. The horizontal line shows the attack success rate when we respectively choose the best $\lambda$ for different blocks.

## H  ADVERSARIAL EXAMPLES SHOW RECONIZABLE FETURES

We insert and fine-tune classifiers (fully connected layer) after each block of ResNet-50. Attack Block1 means attacking the classifier after block1. As show in Fig. 12, the perturbation generated

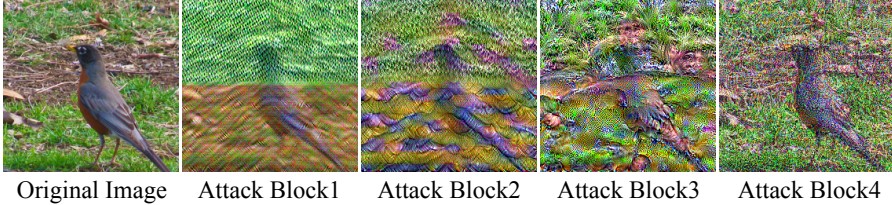

Original Image   Attack Block1   Attack Block2   Attack Block3   Attack Block4

Figure 12: We visualize the targeted adversarial examples by attacking the classifier deployed after the different blocks of normally trained ResNet-50 and by attacking the robust ResNet-50. The perturbation radius $\epsilon$ is $64/255$. The target label is tench. Attack Block1 means generating adversarial example by attacking the classifier deployed after the first block. The Block4 is the last block of ResNet-50 and Attack Block4 means generating adversarial example by attacking the normal ResNet-50.

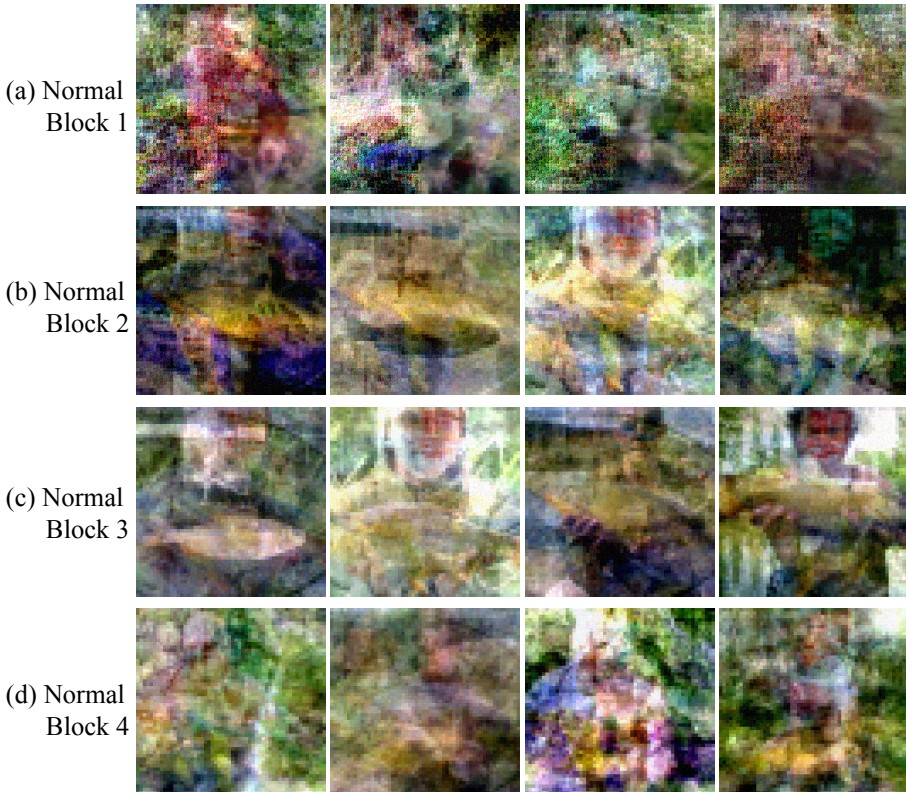

Figure 13: We visualize the images generated by the classifiers fine-tuned after different blocks of a normally trained model. The target label is tench. The early blocks of the model has better generation capability than the last block, which also reflects that the early blocks has learned some distribution-relevant information.

by attacking normal ResNet-50 like unrecognizable noise. However the perturbation generated by attacking the classifiers deployed after block2 and block3 of normal ResNet-50 show recognizable features of tench. This shows that the normal model has also learned some distribution-relevant information.

## I    CONDITIONAL GENERATING EXPERIMENT

We also show the input-gradients in the normal model are somewhat aligned with the gradient of the ground truth data distribution $p_{data}(x|y)$ through a conditional generating experiment using the method proposed by Santurkar et al. (2019). The general ResNet consists of 4 blocks. We insert classifiers (fully connected layers) after each block of ResNet-50 to utilize information from different layers, then fine-tune these classifiers and use these classifiers to generate images. As shown in Fig. 13(given the label tench), the classifier on Block4 (the last block) of the model cannot generate images with recognizable semantic features. The images generated by the classifiers on Block2 and Block3 are not good enough but can be recognized with high confidence. The generative capability of Block1 is poor maybe it does not learn enough class-relevant information. This experiment intuitively shows that manipulating input-gradients can generate images with the features of target distribution (tench).

## J    OTHER METHODS TO ENHANCE THE IMPACT OF EARLY LAYERS

In our paper, **IAA** gives a appealingly simple implementation on ResNet-like neural networks. There are many other ways to adjust the impact of different layers to improve the adversarial transferability. These methods are slightly complex to implement, but they are also effective.

### J.1    TAKING ADVANTAGE OF LOW-LEVEL INFORMATION

In our paper, we find that the later layers may reduce the adversarial transferability and we apply hyper-parameters $\lambda_i$ to reduce the influence of the later layers. We can also use the new loss to enhance the influence of the early layers. We propose a loss to take advantage of low-level information during attacking :

$$\ell_{feature} = \frac{1}{N} \sum_{i=0}^{N-1} KL(feature_{adv}[i], feature_{ori}[i]), \tag{20}$$

where $KL$ is KL divergence, $N$ is the number of blocks in source model, $feature_{ori}$ is the feature of the natural examples extracted by the block $i$, and $feature_{adv}$ is the feature of the adversarial examples extracted by the block $i$. The general ResNet and DenseNet consists of four blocks. We optimize the classification loss and the low-level feature loss jointly during attacking:

$$\underset{\|\delta\|_p \leq \epsilon}{maximize}[\ell(f_\theta(x + \delta), y) + \gamma\ell_{feature}], \tag{21}$$

where $\epsilon$ is a constant of the norm constraint and $\gamma$ is a constant weight for the low-level feature loss. We call this method **Enhancing the Early Blocks (EEB)**. As shown in Fig. 14, we show the influence of the hyper-parameter $\gamma$. Increasing the hyper-parameter $\gamma$ (increasing the weight of low-level information) tends to improve adversarial transferability until it exceeds a threshold, *e.g.*, $\gamma$=100. This may due to $\gamma$ encourages the attack to utilize more transferable low-level information. However, the attack can be insufficient when just using low-level information and ignoring all high-level class-relevant information.

EEB can also improve the adversarial transferability of the deep neural networks without skip connection (VGG16, VGG19). As shown in Tab.

Table 6: Transferability against normal models: the success rates of black-box attacks crafted on VGG16 and VGG19 (DNNs without skip connection). The best results are in bold.

| Model | Attack | RN50 | RN152 | DN121 | DN201 | SE154 | IncV3 | IncV4 | IncRes |
|-------|--------|------|-------|-------|-------|-------|-------|-------|--------|
| VGG16 | PGD | 30.76% | 18.42% | 32.14% | 22.46% | 22.22% | 30.64% | 36.80% | 19.88% |
|  | EEB(ours) | **76.02%** | **59.14%** | **68.74%** | **57.82%** | **64.48%** | **60.74%** | **66.74%** | **41.86%** |
| VGG19 | PGD | 30.84% | 18.40% | 32.20% | 22.46% | 22.26% | 17.36% | 21.68% | 11.74% |
|  | EEB(ours) | **70.26%** | **52.48%** | **70.48%** | **59.02%** | **57.98%** | **43.16%** | **51.66%** | **29.89%** |

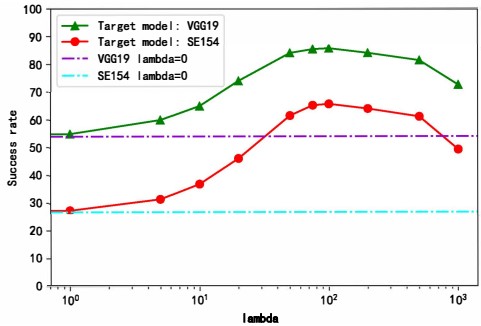

Figure 14: The influence of the hyper-parameter $\lambda$ for the low-level feature loss. The target models are VGG19 and SE154. The source model is RN50. The horizontal lines show the success rate when using normal PGD attack ($\lambda$=0) to generate adversarial examples, and the target models are respectively VGG19 (the purple line) and SE154(the blue line). Optimizing the classification loss and the low-level feature loss jointly during attacking can improve the adversarial transferability.

## J.2 ONLY ATTACK THE EARLY LAYERS

We insert a classifier after the Blocks3 (the penultimate block) of ResNet-50, then fine-tune this classifier, and attack this classifier to generate adversarial examples. Such a simple operation can compete with state-of-the-art methods (SGM). We call this method Dropping the Last Block (DLB).

Table 7: Transferability against different models: the success rates of black-box attacks (untargeted) crafted on ResNet-50 and ResNet-152. The best results are in bold.

| Model | Attack | VGG19 | RN50 | RN152 | DN121 | DN201 | SE154 | IncV3 | IncV4 | IncRes |
|-------|--------|-------|------|-------|-------|-------|-------|-------|-------|--------|
| RN50 | PGD | 45.96% | **99.96%** | 57.44% | 51.62% | 45.7% | 19.7% | 18.96% | 20.96% | 14.30% |
| | MI | 64.86% | 99.92% | 73.22% | 73.50% | 64.33% | 47.20% | 39.08% | 37.35% | 25.26% |
| | DI | 75.06% | 99.96% | 81.65% | 81.98% | 74.80% | 52.42% | 42.58% | 39.30% | 27.12% |
| | SGM | 81.72% | 99.92% | 88.40% | 83.56% | 80.34% | 57.30% | 46.72% | 43.83% | 31.86% |
| | DLB(ours) | 75.18% | 99.94% | 81.84% | 80.42% | 72.12% | 46.06% | 40.14% | 41.60% | 28.08% |
| | EEB(ours) | **92.34%** | 99.94% | **92.05%** | **92.46%** | **88.26%** | **69.08%** | **64.35%** | **60.85%** | **47.77%** |
| RN152 | PGD | 39.32% | 69.72% | 99.96% | 48.44% | 46.00% | 22.20% | 21.50% | 22.58% | 16.72% |
| | MI | 65.42% | 83.40% | 99.92% | 77.60% | 75.79% | 53.00% | 46.50% | 43.32% | 33.08% |
| | DI | 74.01% | 88.18% | 99.96% | 79.46% | 77.81% | 57.49% | 50.28% | 47.16% | 35.10% |
| | SGM | 80.40% | 96.10% | 99.92% | 85.80% | 82.76% | 61.90% | 53.16% | 49.24% | 37.30% |
| | DLB(ours) | 73.94% | 85.84% | 99.94% | 76.74% | 64.30% | 39.62% | 43.72% | 46.84% | 31.56% |
| | EEB(ours) | **90.11%** | **98.46%** | **99.96%** | **93.35%** | **87.26%** | **70.34%** | **63.96%** | **59.06%** | **47.30%** |

## K   IAA UTILIZE THE LOW-LEVEL INFORMATION

Centered kernel alignment (CKA) can reliably identify correspondences between representations in networks (Kornblith et al., 2019; Nguyen et al., 2020). In order to show the difference between natural examples and adversarial examples, we show the CKA similarity between the features of natural examples and adversarial examples extracted by the same normal ResNet-50 (ImageNet). As shown in Fig. 15 (a), the adversarial examples generated by attacking the normal model using PGD show more difference from natural examples in features extracted by deep layers. In Fig. 15 (b), the adversarial example generated by attacking the normal model using our **IAA** are different from the natural example in the shallow layers. For example, in the shallow layers (layer 20 to layer 30), the CKA similarity between the adversarial examples generated by the normal model using PGD and the natural example is about 0.9, while the similarity of the adversarial example generated by **IAA** with the natural example is about 0.7. That is to say, **IAA** uses more low-level information when generating adversarial examples and **IAA** enhances the influence of the shallow layers on generating adversarial perturbation.

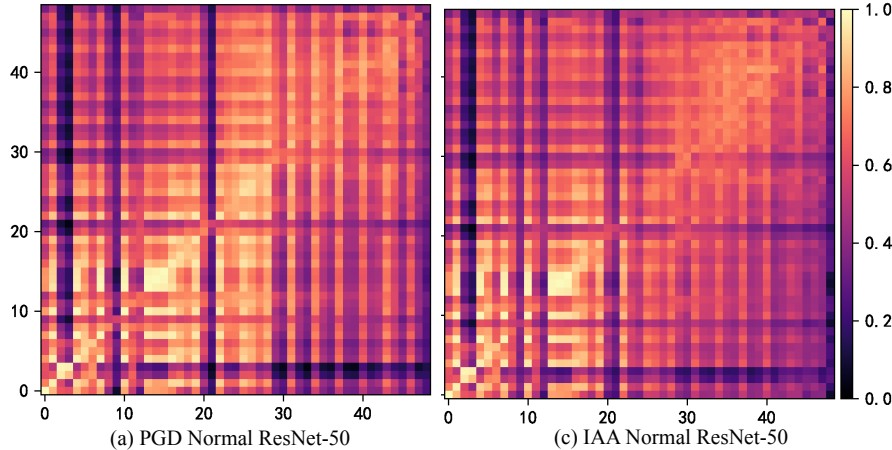

(a) PGD Normal ResNet-50         (c) IAA Normal ResNet-50

Figure 15: The CKA similarity between the features of natural image and adversarial images generated by attacking different models. The horizontal axis represents the features of adversarial examples extracted by different layers of ResNet-50, and the vertical axis represents the features of natural image extracted by different layers of the same ResNet-50. (a) Adversarial examples are generated by attacking normal ResNet-50 with PGD. (b) Adversarial examples are generated by attacking normal ResNet-50 with our **IAA**.

## L    Other corruption cases for the motivation example

As shown in Tab. 8, we added other types of corruption like CIFAR-10-C and ImageNet-C (Hendrycks & Dietterich, 2019), including noise, blur, weather, etc. Consistent with the observations in Fig. 1, the target models are more likely to misclassify corrupted LDD data. We hypothesize that these samples are in the low-density region of the ground truth distribution where both source and target models are not well trained on. Moreover, we find that the adversarial counterparts of LDD have stronger adversarial transferability than that of HDD, which inspires us to match the direction of adversarial perturbation and the direction towards the low-density region to generate adversarial examples with high transferability.

Table 8: The attack success rate when applying different types of corruption on the LDD and HDD against different target models (VGG19, RN50, DN121). The best results are in bold.

| Target | Data | Gaussian noise | Shot noise | Impulse noise | Defocus blur | Glass blur | Zoom blur | Fog | Saturate |
|--------|------|------|------|------|------|------|------|------|------|
| VGG19 | HDD | 6.2% | 6.5% | 6.0% | 8.0% | 7.2% | 14.3% | 6.3% | 6.2% |
|  | LDD | **48.1%** | **46.5%** | **47.5%** | **34.4%** | **31.5%** | **45.9%** | **37.9%** | **37.6%** |
| RN50 | HDD | 4.1% | 4.1% | 1.8% | 4.7% | 4.4% | 9.0% | 6.0% | 6.0% |
|  | LDD | **37.6%** | **36.1%** | **26.2%** | **24.8%** | **23.4%** | **33.8%** | **39.4%** | **34.9%** |
| DN121 | HDD | 1.5% | 1.7% | 1.1% | 3.8% | 3.5% | 7.6% | 3.1% | 1.7% |
|  | LDD | **29.4%** | **27.2%** | **23.5%** | **21.7%** | **20.1%** | **31.4%** | **27.7%** | **24.5%** |

## M    Computation cost

Our proposed method includes two steps. The first step is adjusting the pre-trained source model to match the adversarial attack with the ground truth density decreasing direction by optimizing AAI, and the second step is generating adversarial samples by the modified model similar to the normal PGD attack.

Optimizing AAI to get a strong source model by Bayesian optimization on the pre-trained model needs some extra computation. As shown in Eq. 4, the computation cost for AAI can be reduced to $O(d)$ from $O(d^2)$ by using ideas from sliced score matching (Song et al., 2020), where $d$ is the dimension of data. The computation cost of AAI optimization is 10 hours on Tesla-V100 using ResNet-50.

However, the cost of generating adversarial examples is more important than the cost of adjusting the source model in practice. Once we get the modified model, generating different adversarial

Table 9: Top-1/Top-5 accuracy on different defense models when predicting the black-box adversarial examples crafted by different methods on RN50. We consider two strengths of adversarial perturbation ($\epsilon = 16$ and $\epsilon = 24$). The best results are in bold.

| $\epsilon$ | Attack | NRP | SIN | SIN-IN | AugMix | $\ell_2$ $\epsilon$=0.05 | $\ell_2$ $\epsilon$=0.1 | $\ell_\infty$ $\epsilon$=0.5 | $\ell_\infty$ $\epsilon$=1.0 |
|---|---|---|---|---|---|---|---|---|---|
| | PGD | 1.32/26.48 | 61.66/87.48 | 9.66/56.60 | 51.04/88.00 | 66.06/94.70 | 81.42/97.96 | 94.24/99.40 | 95.82/99.84 |
| 24 | SGM | 0.20/15.86 | 41.50/77.08 | 1.06/31.62 | 15.28/61.20 | 22.68/71.76 | 42.76/84.38 | 82.32/97.48 | 90.82/98.76 |
| | IAA | **0.00/0.02** | **16.18/27.64** | **0.00/0.04** | **0.84/2.44** | **1.82/4.78** | **10.40/19.96** | **63.10/80.18** | **83.42/94.36** |
| | PGD | 3.40/35.78 | 65.56/89.54 | 15.68/64.34 | 60.36/91.90 | 75.62/97.00 | 87.26/98.80 | 96.36/99.60 | 97.12/99.50 |
| 16 | SGM | 1.54/28.98 | 50.28/82.22 | 3.82/43.64 | 27.42/73.00 | 41.88/84.68 | 62.98/93.10 | 90.66/99.24 | 94.60/99.42 |
| | IAA | **0.48/0.98** | **27.76/44.00** | **0.18/0.54** | **5.96/10.98** | **14.08/26.60** | **39.56/59.52** | **84.8/95.1** | **92.98/98.32** |

examples by the modified model shares the same computation with normal PGD adversarial attacks. That is to say, it is very efficient to use our modified model to generate adversarial examples.

## N    ATTACK OTHER DEFENSE METHODS

In Tab. 4, we evaluate that our IAA can fool the robust model by ensemble adversarial training (Tramèr et al., 2018). Moreover, we show that the adversarial examples generated by IAA can also effectively penetrate many other defense methods in Tab. 9. We further add neural representation purifier (NRP) (Naseer et al., 2020) which is a state-of-the-art input processing defense method, and some other robust training methods, including training with AugMix (Hendrycks et al., 2020), Styled ImageNet (SIN) (Geirhos et al., 2019), the mixture of Styled and natural ImageNet (SIN-IN), and adversarial examples (Salman et al., 2020) as defense models. In Tab. 9, we show the top-1 accuracy and top-5 accuracy (ACC1/ACC5) on different defense models when predicting the black-box adversarial examples crafted by different methods. We use the ResNet-50 as the source model. IAA surpasses other attacks in different cases. Our IAA drives the examples to the low-density region by matching the adversarial attack and intrinsic attack. Thus the target models can hardly give a correct prediction on IAA examples, and both the top-1 accuracy and the top-5 accuracy are lower than other methods.

## O    FURTHER DISCUSSION ON DIRECTLY OPTIMIZING THE TRANSFERABILITY.

In our paper, we used the method of optimizing AAI to choose hyperparameters for two reasons. On the one hand, we hope to verify the effectiveness of the metric we proposed. On the other hand, we think that using the transferability for a certain target model to optimize may overfit the target model. Interestingly, Fig.2 and Fig.3 show that the attack success rate among all the target models seems like aligns well under different structural hyperparameters. Maybe it won't be so over-fitted to the target model that it is difficult to select the proper hyperparameters. We conduct experiments on the source model ResNet-50. We use four methods to choose the proper hyperparameters, including using the AAI as the optimization objective ("optimize-AAI"), using the transferability against the target model VGG19 as the optimization objective ("optimize-vgg19"), using transferability against the target model SE154 as the optimization objective ("optimize-senet154"), and using transferability against the ensemble of VGG19 and SE154 as the optimization objective ("optimize-ensemble"). The following table represents the top-1 accuracy and top-5 accuracy (ACC1/ACC5) when predicting the adversarial examples crafted by different methods. Lower accuracy indicates better adversarial transferability. We found that directly using the transferability as the optimization objective can also find proper structural hyperparameters and outperforms the existing methods PGD and SGM. The ensemble-based strategy can slightly improve the adversarial transferability. The computation cost of directly optimizing transferability is half that of optimizing AAI.

Table 10: The attack success rate when using different optimizing methods to choose structural hyperparameters. The best results are in bold.

| | Vgg19 | DN121 | DN201 | SE154 | IncV3 | IncV4 |
|---|---|---|---|---|---|---|
| PGD | 47.00/83.52 | 44.38/84.16 | 46.44/86.92 | 50.44/93.00 | 79.14/96.66 | 78.04/95.68 |
| SGM | 17.28/58.68 | 16.44/63.52 | 19.66/68.46 | 38.7/75.62 | 46.28/84.28 | 50.17/85.95 |
| optimize-AAI | 3.60/5.67 | 2.54/4.37 | 3.36/6.60 | 15.26/18.68 | 23.72/29.36 | 28.62/33.72 |
| optimize-senet154 | 3.96/6.20 | 3.14/4.50 | 4.04/6.34 | **14.60**/18.98 | 21.14/28.86 | 24.72/30.32 |
| optimize-vgg19 | **3.02/4.70** | 2.68/3.94 | 3.78/5.56 | 16.64/20.36 | 23.16/31.36 | 28.28/32.45 |
| optimize-ensemble | 3.20/4.80 | **2.42/3.06** | **3.26/4.70** | 14.86/**17.00** | **19.56/26.86** | **23.30/28.14** |

