# OpenReview forum: "Rethinking Adversarial Transferability from a Data Distribution Perspective"
_ICLR.cc/2022/Conference — ICLR 2022 Poster_

### Official Review · Reviewer_wLmf · 2021-10-31

**Correctness:** 3
**Technical Novelty And Significance:** 3
**Empirical Novelty And Significance:** 2
**Recommendation:** 8
**Confidence:** 4

**Main Review:**

## Strengths
1. The observation on the relationship between low-density data and the high transferability of both random noise and adversarial noise is interesting, likely to inspire future works.
2. Provide theortical analysis on the AAI metric, provide a practical way of computing its value, and emperically shows its high correlation with the attack transferability.
3. Extensive experiments showing the proposed method achieving significantly higher transfer attack performance than SOTA methods, showing the effectiveness of the proposed method.

## Weaknesses
My major concern on this paper is on the practical significance of the proposed AAI metric and IAA algorithm.

1. The necessity of utilizing the AAI metric to generate highly transferable attack is not justified. The paper introduces model hyperparameters $\beta$ and $\lambda$ utilizing the techniques of smoothing activation gradient and use early layer information, which are motivated by the previous works cited in Sec 3.2.1 and 3.2.2, and are not directly related to the AAI metric. As for the Baysian optimization process, since it's a blackbox optimizer assuming no knowledge of the relationship between the parameter and optimization objective, why not directly using the attack transferability as the optimization objective? An ablation study on this is needed to discuss why AAI is needed for this optimization.
2. The cost of evaluating the AAI objective is not discussed. From the formulation in Equation (4) seems like the computation of AAI requires a second-order derivation with respect to x and a sampling process of Gaussian vector v. How is the computation cost of giving an accurate enough estimation of the AAI value comparing to generating an adversarial example with the model?

Besides, there are some other issues in this paper

3. Some steps in the proof of Theorem 1 are hard to understand. For example, how is the $\log$ function got removed from the $\frac{\nabla_x \log p(y|x)}{|| \nabla_x \log p(y|x) ||_2}$ in the first step of the derivation? As the author mentions the proof is similar to the ones from previous works, to make the paper self-contained I would suggest having some more detailed derivation or explaination in Appendix E so that the proof can be understood without referring to your cited papers.
4. Citation should be provided for the baseline models mentioned in the experiment result tables.

**Summary Of The Paper:**

This paper identifies that adversarial examples in the low-density region of the groud truth distribution have much stronger transferability. This observation leads to the AAI metric which evaluate the alignment of the model’s adversarial attack with intrinsic attack direction. The paper further identifies a set of model hyperparameters that can influence the AAI metric, and find the optimal hyperparameter choices to maximize AAI and generate more transferable adversarial examples.

**Summary Of The Review:**

This paper provides interesting theortical insights on how to find highly transferable adversarial examples. However, from the practical side, the proposed training method seems to be inspired from other previous works rather than the observed insight, and whether the discovered AAI metric is practically useful for the optimization process is at doubt. Thus I would suggest a weak reject for now given the potentially low practical significance of the method.

## After rebuttal

After discussing with the author I have a better understanding on the significance of this work. The analysis of AAI motivates the idea of modifying structural hyperparameters of the source model to improve attack transferability, and AAI serves as a good indicator of the transferability towards all model. Although further experiments show it may not be the best objective for choosing the structural hyperparameters, I think it serves as a good theortical strating point for futher analysis of adversarial transferability. Thus I would like to suggest acceptance.

---

> ### Author Response · Authors · 2021-11-13
> **Response to Reviewer wLmf (Part 1)**
>
> We would like to thank you for your thoughtful comments, and we are encouraged that you found our work likely to inspire future works. We hope that our response can address your questions. Please let us know if there are any further questions.
>
> ***
>
> >**Q1.1:** The relationship between the structure hyperparameters ( smoothing hyperparameters $\beta$ and layer hyperparameters $\lambda$ ) and the AAI metric.
>
> **A1.1:** Inspired by the relationship between low-density data and the high transferability of both random noise and adversarial noise, we hypothesize that pushing the data towards the low-density region is an efficient way to improve adversarial transferability. Thus we propose to increase the alignment between the direction of the adversarial perturbation and the direction towards the low-density region (AAI) to generate adversarial examples with high transferability.
>
> In particular, the layer hyperparameter $\lambda$ and the smoothing hyperparameter $\beta$ are included in the structure hyperparameter $\Lambda$ in AAI. Our AAI evaluates the matching of two gradient directions. If there has no smooth guarantee, we cannot calculate it very well (see also our discussion under Theorem 1 in paper). Early layer information helps us to get more distribution-relevant information as the previous work [1] mentioned. Adjusting the structure hyperparameters of the source model can be an appealingly simple method to extract the data distribution information, which is helpful to increase AAI.
>
> ***
>
> >**Q1.2:** The necessity of the AAI metric. Why not directly use the attack transferability as the optimization objective?
>
> **A1.2:** Compared with the existing methods that modify the optimization function for generating adversarial examples to alleviate the over-fitting on the source model, AAI provides a new understanding of this problem from the perspective of data distribution. In this paper, we hypothesize that pushing the data towards the low-density region is an efficient way to improve adversarial transferability.
>
> Based on AAI, we can understand some intriguing phenomena in adversarial transferability. Firstly, the classification accuracy of the source model may not be important for adversarial transferability. As Fig.2 in our paper, we observe that the classifier with low test accuracy but high AAI value can generate adversarial examples with high transferability. This phenomenon verifies the effectiveness of the attack towards the low-density region. Secondly, the top-5 accuracy of the target models on the IAA adversarial examples is significantly lower than other methods. This shows that our adversarial examples are stronger and more intrinsic than other methods. Thirdly, the single-step adversarial perturbation generated by IAA is much stronger than other methods. We think optimizing the AAI contributes to finding the direction to push the sample to the low-density region where different models are hard to make correct predictions.
>
> Guided by AAI, we can design various methods to extract distribution-relevant information in the model and match the alignment between the adversarial attack and intrinsic attack to improve the transferability. In this paper, we found that some structure hyperparameters are useful for extracting distribution-relevant information. We choose these hyperparameters by optimizing AAI in order to show the effectiveness of the AAI metric. Directly optimizing the attack success rate on the target model may be another way to choose hyperparameters. However, choosing which target model to optimize is also a problem, and such an approach is not conducive to understanding the reasons for transferability. Meanwhile, without the help of AAI, it is hard to understand why adjusting these structure hyperparameters with hurting classification accuracy can enhance adversarial transferability. Our AAI is a clear characterization that the key reason for transferability is the attack towards decreasing the ground truth density.
>
> ****

---

> > ### Comment · Reviewer_wLmf · 2021-11-14
> > **Further discussion on your A1.2**
> >
> > I would like to thank the author for the timely response. The response resolves my concerns on the cost of evaluating AAI, and give me better understanding on the significance of the discovered AAI metric in understanding the reasons for better transferability, and in identifying the need of modiffying the structural hyperparameters.
> >
> > Just want to follow up on your A1.2. You mention one issue with directly optimizing the attack success rate on the target model is to choose the target model. Meanwhile, from the observation of Fig 2 and 3 in the paper seems like the attack success rate among all the target models aligns well under different structural hyperparameters. So I suppose choosing the target model isn't a big deal. In this case, if the cost of evaluating attack success rate on a target model is smaller than evaluating the AAI metric, if would be more efficient to directly optimize the attack success rate if it can achieve similar effectiveness.
> >
> > I think it would be the best if the author can provide a quick ablation study on this, say use the transfer success rate on VGG19 as the optimization objective, and see if similar transferability can be achieved on other models, and whether similar hyperparameter can be found as using AAI.

---

> > > ### Author Response · Authors · 2021-11-15
> > > **Further discussion on directly optimizing the transferability**
> > >
> > > We sincerely thank you for the thorough review of our paper and for your further suggestion.
> > >
> > > In our paper, we used the method of optimizing AAI to choose hyperparameters for two reasons. On the one hand, we hope to verify the effectiveness of the metric we proposed. On the other hand, we think that using the transferability for a certain target model to optimize may overfit the target model. Interestingly, Fig.2 and Fig.3 show that the attack success rate among all the target models seems like aligns well under different structural hyperparameters. Maybe it won't be so over-fitted to the target model that it is difficult to select the proper hyperparameters.
> > > We conduct experiments on the source model ResNet-50. We use four methods to choose the proper hyperparameters, including using the AAI as the optimization objective ("optimize-AAI"), using the transferability against the target model VGG19 as the optimization objective ("optimize-vgg19"), using transferability against the target model SE154 as the optimization objective ("optimize-senet154"), and using transferability against the ensemble of VGG19 and SE154 as the optimization objective ("optimize-ensemble"). The following table represents the top-1 accuracy and top-5 accuracy (ACC1/ACC5) when predicting the adversarial examples crafted by different methods. Lower accuracy indicates better adversarial transferability. We found that directly using the transferability as the optimization objective can also find proper structural hyperparameters and outperforms the existing methods PGD and SGM. The ensemble-based strategy can slightly improve the adversarial transferability. The computation cost of directly optimizing transferability is half that of optimizing AAI.
> > >
> > > Thank you again for your interesting suggestions, we will add this part to our paper.
> > >
> > > | Method            | Vgg19       | DN121       | DN201       | SE154       | IncV3       | IncV4       |
> > > |-------------------|-------------|-------------|-------------|-------------|-------------|-------------|
> > > | PGD               | 47.00/83.52 | 44.38/84.16 | 46.44/86.92 | 50.44/93.00 | 79.14/96.66 | 78.04/95.68 |
> > > | SGM               | 17.28/58.68 | 16.44/63.52 | 19.66/68.46 | 38.7/75.62  | 46.28/84.28 | 50.17/85.95 |
> > > | optimize-AAI      | 3.60/5.67   | 2.54/4.37   | 3.36/6.60   | 15.26/18.68 | 23.72/29.36 | 28.62/33.72 |
> > > | optimize-senet154 | 3.96/6.20   | 3.14/4.50   | 4.04/6.34   | **14.60**/18.98 | 21.14/28.86 | 24.72/30.32 |
> > > | optimize-vgg19    | **3.02/4.70**   | 2.68/3.94   | 3.78/5.56   | 16.64/20.36 | 23.16/31.36 | 28.28/32.45 |
> > > | optimize-ensemble | 3.20/4.80   | **2.42/3.06**   | **3.26/4.70**   | 14.86/**17.00** | **19.56/26.86** | **23.30/28.14** |

---

> ### Author Response · Authors · 2021-11-13
> **Response to Reviewer wLmf (Part 2)**
>
>
> >**Q2:** The cost of evaluating the AAI objective is not discussed. How is the computation cost of giving an accurate enough estimation of the AAI value compared to generating an adversarial example with the model?
>
> **A2:** Our proposed method includes two steps. The first step is adjusting the pre-trained source model to match the adversarial attack with the ground truth density decreasing direction by optimizing AAI, and the second step is generating adversarial examples by the modified model similar to the normal PGD attack.
>
> Optimizing AAI to get a strong source model by Bayesian optimization on the pre-trained model needs some extra computation. As shown in Eq. (4), the computation cost for AAI can be reduced to $O(d)$ from $O(d^2)$ by using ideas from sliced score matching [2], where $d$ is the dimension of data. The computation cost of AAI optimization is 10 hours on Tesla-V100 using ResNet-50.
>
> However, we think you are concerned about the cost of generating adversarial examples instead of the cost of adjusting the source model. Once we get the modified model, generating different adversarial examples by the modified model shares the same computation with normal PGD adversarial attacks. That is to say, it is very efficient to use our modified model to generate adversarial examples.
>
> ***
> >**Q3:** Some steps in the proof of Theorem 1 are hard to understand. For example, how is the log function is removed in the first step of the derivation? I would suggest having some more detailed derivation or explanation in Appendix E so that the proof can be understood without referring to your cited papers.
>
> **A3:** Thank you for your suggestion. The $\log$ function can be moved because the normalized $\nabla_x p(y|x)$ and $\nabla_x \log p(y|x)$ are the same direction, as $\frac{\nabla_x \log p(y|x)}{\|\|\nabla_x \log p(y|x)\|\|_2}=\frac{p(y|x)^{-1}\nabla_x p(y|x)}{\|\|p(y|x)^{-1}\nabla_x p(y|x)\|\|_2}=\frac{p(y|x)^{-1}\nabla_x p(y|x)}{p(y|x)^{-1}\|\|\nabla_x p(y|x)\|\|_2}=\frac{\nabla_x p(y|x)}{\|\|\nabla_x p(y|x)\|\|_2}$, where we use the formula: $\nabla_x \log f(x) = f(x)^{-1}\nabla_x f(x)$.
>
> The main tools we use in Eq. (10) are integration by part formula: $\int_{-\infty}^{+\infty} f^{'}(x)g^{'}(x) dx  = \lim_{R\to\infty} f^{'}(x)g(x)|_{-R}^{+R}$
>
>  $-\int_{-\infty}^{+\infty} f^{''}(x)g(x) dx$,  and also the Holder inequality: $|\int_a^bf(x)g(x)dx|\leq [\int_a^b f^2(x)dx]^\frac{1}{2}[\int_a^b g^2(x)dx]^\frac{1}{2}$. We use the property that random vector $v$ is independent of $(x,y)$ in Eq. (11), thus we can switch $\nabla_x$ and $v$.
>
>  We will add the details and explanations in our final version to make our paper self-contained.
>
> ***
>
> >**Q4:** Citation should be provided for the baseline models mentioned in the experiment result tables.
>
> **A4:** Due to space limitations, we provide the citation of the baseline models in Sec. 4.1 and the citation of the baseline methods in Appendix C. We will adjust the layout in the final version with your suggestion.
>
> ***
>
> **References**
>
> [1] Maennel, et al. What Do Neural Networks Learn When Trained With Random Labels?, NeurIPS 2020.
>
> [2] Song, et al. Sliced score matching: A scalable approach to density and score estimation, UAI, 2020.

---

### Official Review · Reviewer_f4GM · 2021-11-02

**Correctness:** 3
**Technical Novelty And Significance:** 3
**Empirical Novelty And Significance:** 3
**Recommendation:** 8
**Confidence:** 4

**Main Review:**

Strengths:
- The proposed method is well-supported by theoretical analysis.
- The experiment is thorough, different scenarios and multiple baseline methods have been covered.
- The results is good. The proposed method outperforms other compared method.

Weaknesses:
- The process of generating adversarial examples includes a bayesian optimization and receptively evaluating the second derivative of the neural network, which could result in high computational cost. What is the computation time of generating adversarial examples? I expect it will slower than normal adversarial attacks but the authors should always report it (including the network structure, hardware information) so that other people could have more understanding on the overall usability of this method.
- I feel there is a disconnection between the motivation and the actual proposed method. In the introduction, the authors show that small Gaussian noise could fool low density data. However, I did not see many connection between this observation and the proposed method. The only transition is that "The most efficient direction towards the low-density region is...". I think the author should elaborate more on this, since this statement is not very obvious and rigorous. Also, what is the formal definition of LDD and HDD?

Minor Comments:
- Since the proposed method can also be used for adversarial attack, I wonder what is the attack success rate of the proposed method? Is it also better than other attacks?
- When citing a paper, one should cite its archival version first instead of non-archival version (e.g., Arxiv, CoRR). For example, citation for (Tramèr et al., 2020) should be "Tramèr, Florian, et al. "Ensemble Adversarial Training: Attacks and Defenses." International Conference on Learning Representations. 2018." instead of "... CoRR, abs/1705.07204, 2020."


**Summary Of The Paper:**

This paper presents a new adversarial attack method which could generate adversarial examples with higher transferability. The proposed method is based on the observation that low-density region of the training data is not well trained. To utilize this, the authors try to align the adversarial direction with the direction to decrease the ground truth density. The proposed method is theoretically support. Their experiments show that the proposed method outperforms multiple adversarial attack method in almost all evaluated scenario. I believe this could be a useful attack method for generating transferable adversarial examples, and providing a strong counterpart for future research on adversarial defense.

**Summary Of The Review:**

Although there are some weaknesses of the paper, based on their experiments, I still believe the proposed method could be useful for generating strong adversarial examples and open the door for future research on defense. I also appreciate their theoretical analysis, although I haven't check the correctness. In general, I am feeling positive on this paper and would like to recommend for acceptance.

==============================
Post rebuttal:

I appreciate the author's comments on my questions. My score will keep the same since it's already reflect my recommendation for acceptance.

---

> ### Author Response · Authors · 2021-11-13
> **Response to Reviewer f4GM**
>
> Thank you for the positive assessment and helpful feedback. We are glad you believe our method provides a strong counterpart for future research on adversarial defense since this is indeed one of our goals. We will address each of your comments in the following.
>
> ****
> >**Q1:** What is the computation time of generating adversarial examples?
>
> **A1:** It is true that optimizing AAI to get a strong source model by Bayesian optimization on the pre-trained model needs some extra computation. As shown in Eq. (4), the computation cost for AAI can be reduced to $O(d)$ from $O(d^2)$ by using ideas from sliced score matching [1], where $d$ is the dimension of data. The computation cost of AAI optimization is 10 hours on Tesla-V100 using ResNet-50.
>
> However, we think you are concerned about the cost of generating adversarial examples instead of the cost of adjusting the source model. Once we get the modified model, generating adversarial examples by the modified model shares the same computation with the normal PGD adversarial attack.
>
> We want to show that we can optimize AAI for a given pre-trained model to further make the adversarial attack from the modified model align with the ground truth density decreasing direction, which is key for strong adversarial transferability. Please feel free to use the modified model to generate adversarial examples, which is as efficient as the widely used PGD adversarial attack.
>
> ****
> >**Q2:** Elaborating more on the connection between the motivation and the proposed method. What is the definition of LDD and HDD?
>
> **A2:** Our motivation example shows that some samples with small Gaussian noise or other corruptions can easily fool many other target models. We hypothesize that these samples are in the low-density region of the ground truth distribution where models are not well trained. LDD (Low-Density Data) and HDD (High-Density Data) are relative concepts in our introduction. Moreover, we find the adversarial counterparts of LDD show high adversarial transferability, which inspires us that the distribution-relevant information can be important in adversarial transferability and reducing the ground truth density of the input may enhance the adversarial transferability.
>
> We theoretically analyze how to match the adversarial attack direction with the ground truth density descent direction. In practice, we found that some structure hyperparameters are useful for extracting distribution-relevant information. We propose to optimize AAI by adjusting the source model's hyperparameters to match the direction of the adversarial attack and the ground truth density descent direction. We experimentally show that pushing the examples towards the low-density region can greatly enhance the adversarial transferability.
>
> ***
>
> **Q3:** Since the proposed method can also be used for adversarial attack, I wonder what is the attack success rate of the proposed method? Is it also better than other attacks?
>
> **A3:** Our paper mainly focuses on the performance of Intrinsic Adversarial Attacks (IAA) in terms of adversarial transferability (black-box). Still, we also think using IAA for the white-box adversarial attacks is interesting and meaningful. It may inspire better defense or other tasks from the distribution perspective. The following table represents the top-1 accuracy and top-5 accuracy (ACC1/ACC5) under different attack strengths when predicting the adversarial examples crafted by different methods. The source model is ResNet-50. IAA has the best performance under different attack strength settings.
>
> | white-box attack | eps=4     | eps=8     | eps=16 | eps=24 |
> |------------------|-----------|-----------|--------|--------|
> | PGD              | 0.08/7.10 | 0.00/2.76    | 0.00/2.18 | 0.00/1.68 |
> | SGM              | 0.10/10.68 | 0.04/3.42 | 0.00/2.38 | 0.00/1.86 |
> | IAA              | **0.08/0.16** | **0.00/0.00**       | **0.00/0.00**    | **0.00/0.00**    |
>
> ****
>
> >**Q4:** When citing a paper, one should cite its archival version first instead of the non-archival version.
>
> **A4:** Thanks very much for pointing out the citation problems. We will rewrite the references in the final version.
>
> ***
>
> **References**
>
>
> [1] Song, et al. Sliced score matching: A scalable approach to density and score estimation, UAI, 2020.

---

### Official Review · Reviewer_cPB9 · 2021-11-07

**Correctness:** 3
**Technical Novelty And Significance:** 3
**Empirical Novelty And Significance:** 3
**Recommendation:** 5
**Confidence:** 4

**Main Review:**

Strength:

- Important topic

- Somehow intuitive and interesting idea

- Sound and feasible solution

- Promising results

Weakness:

- The assumption is intuitive but limited evidence is provided. In particular, the authors only show cases of Gaussian noises, it is unclear to what extent the assumption could be generalized. It would be important to include other noises as well, e.g., corruption cases like CIFAR-10-C.

- Besides robustly trained models, it is unclear how strong the proposed attack could penetrate SOTA defense methods. Please provide more evidence on this.

- It is unclear what is the criterion to decide whether a layer is an early layer or late layer, please justify.

**Summary Of The Paper:**

This paper proposes Intrinsic Adversarial Attack (IAA), a transfer attack method based by jointly matching data distribution. The key assumption of this paper is that the DNN might not be well trained on low-density regions (LDD). Therefore, taking data distribution into consideration during attack could potentially improve the transferability. Empirical evaluation on different model architectures, normal models, robustly trained models, and ensemble-based attack context, demonstrates the advantage of IAA.

**Summary Of The Review:**

Overall, this paper is well written and works on an important problem. The assumption is simple and intuitive, the proposed method is also feasible and sound.

Even though, there are still quite a few concerns. I am on the borderline of this paper and could be easily flipped to the other side based on the authors' response.

---

> ### Author Response · Authors · 2021-11-13
> **Response to Reviewer cPB9 (Part 1)**
>
> We sincerely appreciate your valuable comments, and we are encouraged that you found our work interesting and important. We hope that our response can address your questions. Please let us know if there are any further questions.
> ****
>
> >**Q1:** The assumption is intuitive but limited evidence is provided. It would be important to include other noises as well, e.g., corruption cases like CIFAR-10-C.
>
> **A1:** Great suggestion! As shown in the following table, we added other types of corruption like CIFAR-10-C and ImageNet-C [1], including noise, blur, weather, etc. Consistent with the observations in our paper, the target models are more likely to misclassify corrupted LDD data. We hypothesize that these samples are in the low-density region of the ground truth distribution where both source and target models are not well trained on. Moreover, we find that the adversarial counterparts of LDD have stronger adversarial transferability than that of HDD, which inspires us to match the direction of adversarial perturbation and the direction towards the low-density region to generate adversarial examples with high transferability.
>
> | Target model | Data | Gaussian noise | Shot noise | Impulse noise | Defocus_blur | Glass_blur | Zoom_blur | Fog  | Saturate |
> |--------------|------|----------------|------------|---------------|--------------|------------|-----------|------|----------|
> | VGG19        | HDD  | 6.2            | 6.5        | 6             | 8            | 7.2        | 14.3      | 6.3  | 6.2      |
> | VGG19        | LDD  | **48.1**           | **46.5**       | **47.45**         | **34.4**         | **31.5**       | **45.9**     | **37.9** | **37.6**     |
> | RN50         | HDD  | 4.1            | 4.1        | 1.8           | 4.7          | 4.4        | 9         | 6    | 6        |
> | RN50         | LDD  | **37.6**           | **36.1**       | **26.2**          | **24.8**         | **23.4**       | **33.8**      | **39.4** | **34.9**     |
> | DN121        | HDD  | 1.5            | 1.7        | 1.1           | 3.8          | 3.5        | 7.6       | 3.1  | 1.7      |
> | DN121        | LDD  | **29.4**           | **27.2**       | **23.5**          | **21.7**         | **20.1**       | **31.4**      | **27.7** | **24.5**     |
>
> ****
>
> >**Q2:** Besides adversarially trained models, it is unclear how strong the proposed attack could penetrate SOTA defense methods.
>
> **A2:** Thanks for the suggestions on experiments about more defense methods. We have considered 3 robustly trained models using ensemble adversarial training [2] in our paper. Here we provide more experiments to show that the adversarial examples generated by IAA can also effectively penetrate many other defense methods. We further add neural representation purifier (NRP) [3] which is a state-of-the-art input processing defense method, and some other robust training methods, including training with Augmix [5], Styled ImageNet (SIN) [4], the mixture of Styled and natural ImageNet (SIN-IN), and adversarial examples ($\ell_2$ and $\ell_\infty$) [6] as defense models. In the following table, we show the top-1 accuracy and top-5 accuracy (ACC1/ACC5) on different defense models when predicting the black-box adversarial examples crafted by different methods. We use the ResNet-50 as the source model. IAA surpasses other attacks in different cases. Our IAA drives the examples to the low-density region by matching the adversarial attack and intrinsic attack. Thus the target models can hardly give a proper prediction on IAA examples, and both the top-1 accuracy and the top-5 accuracy are lower than other methods.
>
> | Attack     | NRP        | SIN         | SIN-IN      | Augmix     | $\ell_2$ $\epsilon$=0.05      | $\ell_2$ $\epsilon$=0.1      | $\ell_\infty$ $\epsilon$=0.5     | $\ell_\infty$ $\epsilon$=1      |
> |------------|------------|-------------|-------------|------------|-------------|-------------|-------------|-------------|
> | PGD eps=24 | 1.32/26.48 | 61.66/87.48 | 9.66/56.60  | 51.04/88   | 66.06/94.70 | 81.42/97.96 | 94.24/99.40 | 95.82/99.84 |
> | SGM eps=24 | 0.20/15.86 | 41.50/77.08 | 1.06/31.62  | 15.28/61.20 | 22.68/71.76 | 42.76/84.38 | 82.32/97.48 | 90.82/98.76 |
> | IAA eps=24 | **0.00/0.02**     | **16.18/27.64** | **0.00/0.04**      | **0.84/2.44**  | **1.82/4.78**   | **10.40/19.96**  | **63.10/80.18**  | **83.42/94.36** |
> | PGD eps=16 | 3.40/35.78 | 65.56/89.54 | 15.68/64.34 | 60.36/91.90 | 75.62/97.00    | 87.26/98.80 | 96.36/99.60  | 97.12/99.50 |
> | SGM eps=16 | 1.54/28.98 | 50.28/82.22 | 3.82/43.64  | 27.42/73.00   | 41.88/84.68 | 62.98/93.10 | 90.66/99.24 | 94.60/99.42 |
> | IAA eps=16 | **0.48/0.98**  | **27.76/44.00**    | **0.18/0.54**   | **5.96/10.98** | **14.08/26.60** | **39.56/59.52** | **84.80/95.10**   | **92.98/98.32** |
>
> ****

---

> ### Author Response · Authors · 2021-11-13
> **Response to Reviewer cPB9 (Part 2)**
>
>
> >**Q3:** It is unclear what is the criterion to decide whether a layer is an early layer or late layer, please justify.
>
> **A3:** As for the early layers and the late layers, we have borrowed the concepts from [7] and they are relative concepts. The early layers refer to the layers close to the input, and the late layers refer to the layers close to the output.
>
>
> ****
>
> **References**
>
> [1] Hendrycks, et al., Benchmarking Neural Network Robustness to Common Corruptions and Perturbations, ICLR 2018.
>
> [2] Tramèr, et al. Ensemble Adversarial Training: Attacks and Defenses, ICLR 2018.
>
> [3] Naseer, et al. A self-supervised approach for adversarial robustness, CVPR 2020.
>
> [4] Geirhos, et al. Imagenet-trained CNNs are biased towards texture; increasing shape bias improves accuracy and robustness, ICLR 2019.
>
> [5] Hendrycks, et al. AugMix: A Simple Data Processing Method to Improve Robustness and Uncertainty, ICLR 2020.
>
> [6] Salman, et al. Do adversarially robust imagenet models transfer better?, NeurIPS 2020.
>
> [7] Mehrer, et al. Individual differences among deep neural network models, Nature communications 11.1 (2020).

---

### Author Response · Authors · 2021-11-19
**Update Manuscript**

We would like to thank all of the reviewers again for helping us improve the paper. We uploaded a revised version of our paper and marked the major modifications in blue for visibility. In short,

1. We add other types of corruption like CIFAR-10-C and ImageNet-C for our motivation example in Appendix L.

2. We provide more experiments to show that the adversarial examples generated by IAA can also effectively penetrate many other defense methods in Appendix M.

3. We add a discussion of computation cost in Appendix N.

4. We revise the citation problem.

5. We add the details for Theorem 1 in Appendix E.

6. We add an ablation study that directly optimizes transferability to find our structural hyperparameters in Appendix O.

Thank you all again for your precious and insightful suggestions. Please let us know if you have additional questions or ideas for improvement.

Kind regards, Authors

---

### Public Comment · ~Zhengyu_Zhao1 · 2022-02-07
**Close related work and Attacks with more iterations**

Dear Authors,

Congratulations on the acceptance of this interesting work!

Here are two very close (workshop) papers published in 2021 that also showed that using softplus function [1] and removing residual blocks [2] lead to much better adversarial transferability.
Since you already showed that optimizing directly with transferability on a surrogate target model as the objective can lead to the same good results, does this mean that IAA is not necessary (but actually softplus function+removing residual blocks play the role here)?

I have also noticed that all attacks in your paper have been only tested with 10 iterations. However, it has been shown by recent work that such a low number of iterations may cause misleading evaluation: many attacks can not converge well with 10 iterations, especially for the targeted goal [3].

Looking forward to your reply!

[1] https://phibenz.github.io/publication/backpropagating_smoothly/

[2] https://phibenz.github.io/publication/stochastic_depth/

[3] Zhao et al. “On Success and Simplicity: A Second Look at Transferable Targeted Attacks”. NeurIPS 2021

---

> ### Public Comment · ~Yao_Zhu2 · 2022-02-15
> **Response to Zhengyu**
>
> Hi Zhengyu,
>
> Thanks for your interest in our paper. We focus on providing a new perspective to further understand adversarial transferability. As we discussed with the reviewer wLmf, the method in our paper is somewhat close to the previous works cited in Sec 3.2.1 and 3.2.2 and the paper you point out.
> These related methods are effective in practice, but it is not clear why they work. AAI is a new metric to evaluate the alignment of adversarial direction and the ground truth density decreasing direction. We hypothesis that adversarial examples in the low-density region are much more transferable. The analysis of AAI motivates our idea of modifying structural hyperparameters of the source model to improve attack transferability, and our implementation is different from previous research. We only modify a few structure hyper-parameters rather than retrain/fine-tune the model.
>
> Thanks for pointing out your paper! We follow the same standard experiment setting as baselines compared in our paper.
> Our paper shows that our method performs the best in the one-step attack and multi-step attack (10 steps), which reveals that our method can effectively find the effective attack direction. We think this is because our adversarial perturbation drives the examples towards the low-density region.
> After reading your paper, we find it is interesting that more steps can improve the transferability of the targeted attack. Thanks for your suggestion that considering more iterations during the attack, we will take it into consideration in our future work.

---

> > ### Public Comment · ~Zhengyu_Zhao1 · 2022-02-16
> > **Thanks and remaining concerns**
> >
> > Thanks for your clarification and your interest in our work [3].
> >
> > I still have concerns about the necessity of the IAA. Based on the experimental results (cf. appendix O), the transferability improvement is not relevant to the IAA objective at all, but purely benefitting from the use of smoothing (softplus) function and removing later layers. Considering that directly using a transferability objective on a (surrogate) target model is more efficient than using IAA and not suffering from the overfitting problem, what is the motivation to still talk about IAA in the main paper? Will the data distribution assumption still hold given this finding?
> >
> > Looking forward to your reply!

---

> > > ### Public Comment · ~Yao_Zhu2 · 2022-02-21
> > > **Response to Zhengyu**
> > >
> > > Thanks for your further discussion. We appreciate your interest in our work.
> > >
> > > We hope to provide new insight into understanding adversarial transferability from a distribution perspective. AAI can be used to approximately evaluate the alignment of the models' gradient and the gradient of the ground truth distribution, and we aim to extract the distribution-relevant information, which helps to reduce the dependence on the source model (As shown in Fig.9 and Fig.10, the perturbation generated by different models using IAA have stronger correction than other methods). If there has no smooth guarantee, we cannot calculate AAI very well (see also our discussion under Theorem 1 in paper). Early layer information helps us to get more distribution-relevant information as the previous work in 3.2.2 mentioned. This is why we chose these structure hyperparameters used in this paper. As shown in Fig.2 and Fig.3, the model with better AAI shows stronger adversarial transferability.
> > >
> > > Moreover, our method is different from the existing methods, such as removing the later layers (just consider the early layers) or using more gradients from the skip connections (just consider the backward process). Guided by AAI, we think that the ground truth distribution-relevant information is much more important than test accuracy (Fig.2). We directly adjust the structure hyperparameters in the source model, which affects both the forward process and the backward process (Eq.(8)). Our method reduces the test accuracy but is helpful to increase the AAI and greatly enhance the adversarial transferability.
> > >
> > > It's true that we can directly optimize the hyperparameters by a target model, as we showed in our appendix O. This may help us to get good transferability, but why it works so well remains unknown. However, AAI can guide us to choose these hyper-parameters and provide an effective explanation for why this method work which is the main focus of our work.

---

> > > > ### Public Comment · ~Zhengyu_Zhao1 · 2022-02-21
> > > > **Thank you!**
> > > >
> > > > Thanks for your further detailed clarification. Looking forward to (your) follow-up work on related topics!

---

### Public Comment · ~Jiaming_Zhang1 · 2023-03-28
**Do you plan to open the code?**

Dear authors,

It seems I can't find the code in the paper or on GitHub, do you plan to release it?

---

> ### Public Comment · ~Yao_Zhu2 · 2023-03-28
> **Thanks for your comment**
>
> Dear Jiaming,
>
> Our code (IAA) can be found in the work (https://arxiv.org/abs/2211.09565):
>
> https://github.com/ZhengyuZhao/TransferAttackEval/tree/main/attacks
>
> If you have any further questions, please do not hesitate to email me.
>
> Best regards,
>
> Yao Zhu

---

### Decision · Program_Chairs · 2022-01-20

**Decision:**

Accept (Poster)

**Comment:**

This work studied an important issue, i.e., adversarial transferability, in adversarial examples. It provides a novel perspective that samples in  the low-density region of the ground truth distribution where models are not well trained have stronger transferability across different models. Based on that, it proposed a metric called Alignment between its Adversarial attack and the Intrinsic attack (AAI) to indicate transferability. Inspired by the connection between AAI and transferability, this work further proposed to replace the regular ReLU activation with some smooth activation functions, to enhance the transferability.

Most reviewers appreciate that the observation is interesting, and the theoretical analysis and the proposed method are intuitive. The reviewers posed some important comments on experiments, and the relationship between the proposed method and the proposed metric. The authors provided satisfied responses to most of these concerns. Although there is one remaining concern that AAI may be not the best metric to choose the structural hyper-parameters, the reviewer still thought it is a good theoretical starting point to further analyze the adversarial transferability.

After reading the submission, reviewers' comments and the discussions between reviewers and authors, I believe that this work has provided a valuable perspective, a reasonable theoretical analysis and an effective solution for adversarial transferability. It could inspire further studies on adversarial transferability.